# Bridging Degradation Discrimination and Generation for Universal Image Restoration

**JiaKui Hu**[1,2,3]**, Zhengjian Yao**[1,2,3]**, Lujia Jin**[4]**, Yanye Lu**[1,2,3*]
[1]Institute of Medical Technology, Peking University
[2]Biomedical Engineering Department, College of Future Technology, Peking University
[3]National Biomedical Imaging Center, Peking University
[4]China Mobile Research Institute

## Abstract

Universal image restoration is a critical task in low-level vision, requiring the model to remove various degradations from low-quality images to produce clean images with rich detail. The challenges lie in sampling the distribution of high-quality images and adjusting the outputs on the basis of the degradation. This paper presents a novel approach, Bridging Degradation discrimination and Generation (BDG), which aims to address these challenges concurrently. First, we propose the Multi-Angle and multi-Scale Gray Level Co-occurrence Matrix (MAS-GLCM) and demonstrate its effectiveness in performing fine-grained discrimination of degradation types and levels. Subsequently, we divide the diffusion training process into three distinct stages: generation, bridging, and restoration. The objective is to preserve the diffusion model's capability of restoring rich textures while simultaneously integrating the discriminative information from the MAS-GLCM into the restoration process. This enhances its proficiency in addressing multi-task and multi-degraded scenarios. Without changing the architecture, BDG achieves significant performance gains in all-in-one restoration and real-world super-resolution tasks, primarily evidenced by substantial improvements in fidelity without compromising perceptual quality.

## 1 Introduction

Image restoration aims to remove degradations from low-quality (LQ) images and to reconstruct high-quality (HQ) images while maintaining consistent semantic and texture details. In the context of deep learning (LeCun et al., 2015), image restoration can be further conceptualized as a conditional generation task: employing LQ images as a condition, using neural networks to sample the distribution of the corresponding HQ images.

Universal image restoration (Luo et al., 2023; Zheng et al., 2024; Hu et al., 2025a) represents an emerging and significant subfield of image restoration, which is intended to challenge the restoration model to effectively identify a myriad of complex or previously unseen degradations. This requires that the restoration model possesses two capabilities: *(1) degradation discrimination* and *(2) conditional generation*. The former propels the model to discern the degradation present in input images, thereby enhancing the model's adaptability, whereas the latter enables the model to generate the HQ images based on the LQ images, fulfilling the restoration. These two capabilities lead researchers to develop universal image restoration models from two distinct perspectives. Several methods (Li et al., 2022a; Zhang et al., 2023a; Potlapalli et al., 2023; Marcos V. Conde, 2024; Hu et al., 2025a) employ additional degradation discrimination networks (or parameters) to guide the model in identifying the degradation. This approach has demonstrated efficacy in all-in-one image restoration tasks. However, it results in over-smoothed outcomes due to the fidelity-focused learning objectives, and may not perform well in real-world scenarios. In contrast, Wang et al. (2024); Yu et al. (2024); Wu et al. (2024); Lin et al. (2024); Chen et al. (2025) focus on effectively exploiting generation prior of the pre-trained generation model to output rich textures. These methods have

---

*Corresponding author.

proven effective in real-world and photo-realistic image restoration tasks. However, for the all-in-one image restoration task, these methods struggle to produce detailed content consistent with the LQ image. This issue potentially arises from the diffusion model erroneously interpreting mildly degraded images as severely degraded, compelling it to generate rich but inconsistent detail, thereby causing less fidelity.

To maintain the generation prior of the diffusion model while improving its restoration fidelity in other common tasks, we propose **B**ridging **D**egradation discrimination and **G**eneration (**BDG**). The essence of BDG lies in the seamless integration of fine-grained degradation discrimination capabilities with robust high-quality image generation within a single model. This configuration enables the model to effectively address degradation presented in diversified or complex forms and subsequently produce HQ images.

For degradation discrimination, we employ **M**ulti-**A**ngle and multi-**S**cale **G**ray **L**evel **C**o-occurrence **M**atrix (**MAS-GLCM**) to distinctly identify complex and diversified degradations. Through visualization, T-SNE clustering, and KNN classification, we practically demonstrate that our MAS-GLCM surpasses previous degradation characterizations, *e.g.*, gradients (Ma et al., 2020), frequency (Ji et al., 2021), parameters (Potlapalli et al., 2023), and instructions (Luo et al., 2023; Marcos V. Conde, 2024), in advanced fine-grained degradation discrimination. Based on this finding, BDG utilizes MAS-GLCM to endow the model with degradation discrimination abilities.

For bridging degradation discrimination and generation prior, we design a three-stage diffusion method by modifying the parameters in the diffusion reserve formula. **(I)** During the generation stage, the model incrementally captures pixel dependency through a denoising process. **(II)** In the bridging stage, the model incorporates residual information as an input condition. The inherent degradation discrimination capacity of the residual (Tang et al., 2024a) provides advantageous conditions for the introduction of a fine-grained degradation discrimination ability. Accordingly, we accomplish BDG by aligning the GLCM features with the diffusion features in the bridging stage. **(III)** In the restoration stage, the focus of the model shifts from generating HQ images to prioritizing training in restoration ability. During this stage, continued alignment of the GLCM features with the diffusion features is necessary. This alignment ensures that the model's fine-grained degradation discrimination ability can be retained. After firmly bridging the degradation discrimination and the generation prior, we attain superior restoration performance in both all-in-one image restoration and real-world super-resolution, thereby illustrating BDG's effectiveness.

The proposed BDG framework facilitates the precise attainment of a high-fidelity, universal restoration model that effectively accommodates arbitrary degradation arising from the image generation model. By capitalizing on the nuanced degradation discrimination capabilities of MAS-GLCM, coupled with the integration of a robust pre-trained generative model, models trained within the BDG paradigm achieve an optimal equilibrium between content fidelity and detail restoration in the context of image restoration tasks.

## 2 Related work

**Degradation diversities in restoration.** To enhance the adaptability of restoration models, researchers have initiated studies that aim to develop a single model capable of addressing multiple restoration tasks, a process known as all-in-one restoration (Li et al., 2022a). In this context, the restoration model is expected to effectively restore input images with various degradations. Numerous methods (Li et al., 2022a; Potlapalli et al., 2023; Zhang et al., 2023a; Luo et al., 2023; Marcos V. Conde, 2024; Hu et al., 2025a;c) are designed to improve all-in-one restoration performance by introducing the degradation discrimination capability. AirNet (Li et al., 2022a) uses MoCo (He et al., 2020), while IDR (Zhang et al., 2023a) creates physical degradation models for identifying degradations. PromptIR (Potlapalli et al., 2023) incorporates additional parameters through dynamic convolutions to enable universal image restoration without relying on embedded features. DCPT (Hu et al., 2025a) approaches the restoration model as a degradation classifier to encourage it to learn about the diversity of degradation. In contrast, (Wang et al., 2023b; Zheng et al., 2024; Qin et al., 2024) aim to allow the model to extract features independently of degradation, ensuring that its output is solely linked to the intrinsic distribution of the images.

**Generation priors for real-world restoration.** To improve the applicability of restoration models in real-world settings (Wang et al.), researchers have begun to incorporate generation priors into these models. Pre-trained in high-quality real-world images, large-scale image generation models (Esser et al., 2021; Rombach et al., 2022; Peebles & Xie, 2023; Tian et al., 2024; Esser et al., 2024; Liu et al., 2024) are considered proficient in fitting complex image distributions. Existing real-world restoration techniques (Kawar et al., 2022; Fei et al., 2023; Wang et al., 2024; Wu et al., 2024; Yu et al., 2024; Lin et al., 2024; Yao et al., 2026) attempt to leverage this capability of pre-trained large-scale image generation models to address scenarios involving intricate image distribution challenges. DDRM (Kawar et al., 2022) is the pioneering method for employing the generative prior of a diffusion model (Ho et al., 2020), thus markedly enhancing the perceived effectiveness of the restoration model. GDP (Fei et al., 2023) attributes these improvements offered by pre-trained generative models to their inherent general image priors. StableSR (Wang et al., 2024) capitalizes on the generative prior of stable diffusion (Rombach et al., 2022), leading to a substantial enhancement in the perceived effectiveness of the restoration model in real-world scenarios. DiffBIR (Lin et al., 2024) expands this capability to include a variety of blind image restoration tasks. SUPIR (Yu et al., 2024) advances the field by making significant contributions to large-scale restoration models through the integration of scaling. SeeSR (Wu et al., 2024) explores how high-level semantics can better assist diffusion-based restoration. PURE (Wei et al., 2025) also successfully used pre-trained autoregressive MLLM (Hu et al., 2025d;b) to adapt to real-world super-resolution.

## 3 METHODS

### 3.1 DEGRADATION CHARACTERIZATION

Methods utilizing degradation characterizations Ma et al. (2020); Ji et al. (2021); Potlapalli et al. (2023); Luo et al. (2023); Marcos V. Conde (2024) have been widely demonstrated to enhance restoration performance under various degradations. Existing degradation characterizations are tied to image content, *e.g.*, the Sobel operator (Ma et al., 2020) focuses the texture at edge. When used to align with the restoration network, the network predominantly aligns with image content, thus inhibiting the ability to capture degradation-specific information.

To achieve a more refined degradation characterization that is minimally affected by image content, we introduce the **MAS-GLCM** and substantiate its proficiency in discriminating degradation. MAS-GLCM is designed on the basis of GLCM, which serves as an effective extractor of image texture, depicting the texture characteristics of an image by evaluating spatial associations between pixels at different gray levels. Specifically, the GLCM constitutes a matrix that computes the frequency with which pixel pairs of given gray levels co-occur within an image. Each matrix element quantifies the occurrences of one gray value in conjunction with another at specified distances and orientations. Since its computation does not involve processing the image's content, GLCM's result inherently discards information about the image content. Given a gray image $I \in \mathbb{R}^{H \times W}$, its GLCM $M$ can be expressed as follows.

$$M_{\Delta x, \Delta y}(i,j) = \sum_{x=1}^{W} \sum_{y=1}^{H} \begin{cases} 1, & \text{if } \begin{aligned} &I(x,y) = i \text{ and} \\ &I(x+\Delta x, y+\Delta y) = j \end{aligned} \\ 0, & \text{otherwise} \end{cases} \tag{1}$$

where $\Delta x, \Delta y$ denotes the distances selected in the horizontal and vertical directions.

**MAS-GLCM.** Our objective is to determine the average value of GLCM in various groups of $\Delta x, \Delta y$, to fully extract information that encompasses multiple angles and scales, thereby extracting degradation-related information at multiscales and avoiding being trapped in locality. Considering $\Delta x, \Delta y$ can be formulated by the angle $\vartheta$ and the module $l$ as $\Delta x = l \cdot sin(\vartheta), \Delta y = l \cdot cos(\vartheta)$. Given multiple $\vartheta$ and $l$, we compute the average value of their $M$ to obtain our MAS-GLCM $M_{mas}$, which can be formulated as follows.

$$M_{mas} = \frac{1}{n \times m} \sum_{i=1,j=1}^{L,\Theta} M_{L_i \cdot sin(\Theta_j), L_i \cdot cos(\Theta_j)}, \tag{2}$$

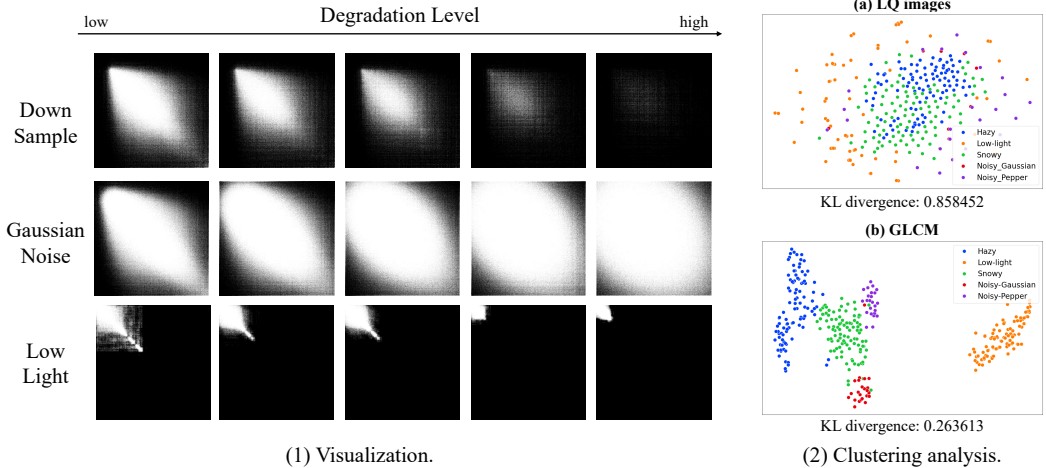

Figure 1: (1) Visualization of MAS-GLCM in varying degradation levels. With an increase in degradation levels, the MAS-GLCM exhibits significant transformations. (2) The results of the T-SNE analysis for LQ images and MAS-GLCM across various degradation types demonstrate that MAS-GLCM possesses an enhanced capacity to distinguish between degradation types.

where $L$ represents different scales $L = l_1, l_2, \cdots, l_n$ and $\Theta$ represents different angles $\Theta = \vartheta_1, \vartheta_2, \cdots, \vartheta_m$.

**Fine-grained degradation discrimination.** We demonstrate that $M_{mas}$ has fine-grained degradation discrimination capability through visualization and clustering analysis, as shown in Figure 1.

- **Visualization.** Figure 1 (1) displays the MAS-GLCM at various degradation levels. We select a clean image with three simulated degradation types: downsampling, Gaussian noise, and low-light. For each degradation type, five distinct levels are defined, details in Appendix A. The visualization indicates that MAS-GLCM significantly changes with varying degradation levels.

- **Clustering analysis.** Figure 1 (2) illustrates the T-SNE results associated with the degradation cluster using LQ images and MAS-GLCM. Following T-SNE clustering, MAS-GLCM exhibits a much lower KL divergence compared to LQ images, indicating superior clustering efficacy. More results on real-world datasets are shown in Appendix A. This claim is further supported by the visualization of the clustering results.

- **Classification analysis.** Table 1 compares the degradation characteristics in classification tasks for type and level, including haze, low light, snow, Gaussian noise, and Pepper noise, with Gaussian noise variances of 15, 25, 50, 75, and 100, using a KNN classifier (details in Appendix A). The classification results demonstrate that MAS-GLCM shows superior performance, especially in the fine-grained degradation level classification task.

| Degradation Characterization | Type Acc (%) | Level Acc (%) |
|---|---|---|
| LQ images | 51.44 | 20.00 |
| Sobel (gradient) | 40.80 | 23.33 |
| Laplace (gradient) | 83.05 | 20.83 |
| Fourier | 65.80 | 30.83 |
| **MAS-GLCM (Ours)** | **97.13** | **74.17** |

Table 1: MAS-GLCM has substantial capability in the classification of both types and levels of degradation.

## 3.2 CONDITIONAL GENERATION

The diffusion model has been widely proven to have superior generative capacities (Rombach et al., 2022). Their inherent generation ability (Fei et al., 2023) significantly helps the model in addressing restoration tasks in real-world degradation. Following (Wang et al., 2024; Zheng et al., 2024; Wu et al., 2024), we use the diffusion model to learn how to fit the distribution of HQ images.

**Condition mechanisms.** In contrast to image generation tasks, image restoration tasks possess a pronounced condition in the form of the LQ image, which serves as a guidance for the model. Common techniques for incorporating such conditional information in diffusion-based image restoration models include methods such as Cross Attention, ControlNet (Zhang et al., 2023b), residuals (Yue et al., 2023; Liu et al., 2024), etc. Following (Liu et al., 2024; Zheng et al., 2024), we use residuals and LQ images or their latents as conditions.

In a nutshell, the forward process of the diffusion model we use is as follows.

$$x_t = x_{t-1} + \alpha_t x_{res} + \beta_t \epsilon_{t-1} - \delta_t x_{lq}, \tag{3}$$

where $x_t$ is the diffusing result in timestep $t$, $x_{res} := x_{lq} - x_{hq}$ is the residual of the LQ image (or its latent) $x_{lq}$ and the HQ image (or its latents) $x_{hq}$. $\alpha_t$, $\beta_t$, and $\delta_t$ is the noise coefficient of $x_{res}$, standard Gaussian noise $\epsilon$, and $x_{lq}$, respectively.

In the sampling process, we omit the noise term to change this diffusion to an implicit probabilistic model (Mohamed & Lakshminarayanan, 2016). The derivation can be found in Appendix B.

$$x_{t-1} = x_t - \alpha_t x^\theta_{res} - \frac{\beta_t^2}{\bar{\beta}_t} \epsilon^\theta + \delta_t x_{lq}. \tag{4}$$

**Discussion.** Eq. 4 is the basis for us to bridge the degradation discrimination and the generative prior. As derived from Eq. 4, the image generation or restoration capability of this diffusion model is governed by three parameters: $\alpha$, $\beta$, and $\delta$. We shall examine the performance of this diffusion model in the subsequent three cases:

1. **Generation.** $\alpha_t \equiv 0$ and $\delta_t \equiv 0$, Eq. 4 will degenerate into $x_{t-1} = x_t - \frac{\beta_t^2}{\bar{\beta}_t} \epsilon^\theta$. This formula is formally equivalent to the denoising formula of the Variance Exploding (VE) SDE (Song et al., 2021). The model in this stage only has generation abilities, as it has not been provided with the necessary conditions for restoration, such as $x_{res}$ or $x_{lq}$.

2. **Bridging.** Only $\delta_t \equiv 0$, Eq. 4 will degenerate into $x_{t-1} = x_t - \alpha_t x^\theta_{res} - \frac{\beta_t^2}{\bar{\beta}_t} \epsilon^\theta$. The diffusion model is capable of comprehending degradations by leveraging the degradation-aware information (residual $x_{res}$ (Tang et al., 2024b)), while preserving its generation prior.

3. **Restoration.** All parameters are scheduled as normal, Eq. 4 does not degenerate. Note that we set $\alpha_t \neq \delta_t$, so the introduction of $x_{lq}$ will not be diluted by $x_{res}$. Due to direct injection of $x_{lq}$, the predicted image $x_0^\theta$ can have stronger fidelity. *Different from the DiffUIR* (Zheng et al., 2024), sampling of BDG predicts both noise and residual to obtain the generation (noise prediction) prior, whereas DiffUIR only predicts residual, missing out on acquiring the generation prior in diffusion models.

### 3.3 TRAINING

The BDG training phase can be divided into three distinct stages, each corresponding to the cases previously discussed. This three-stage methodology is intended to enable the restoration model not only to retain the generation model's capability for recovering detailed textures, but also to acquire enhanced knowledge pertinent to degradation. This approach is designed to improve the model's adaptability to varying task requirements and degradation scenarios. Figure 2 clearly shows this training process. We introduce each stage sequentially.

**Generation Pre-training.** We set the coefficients $\alpha_t \equiv 0$ and $\delta_t \equiv 0$ to correspond to the generation stage of the diffusion model. At this stage, the model mainly assimilates the generation prior from extensive and high-quality image datasets.

**Bridging stage.** We maintain $\delta_t \equiv 0$ in this stage. A primary objective of this stage is to preserve the model's generation capabilities. Specifically, conditional on Eq. 4, it is imperative that the model accurately predicted the distribution $p_\theta(x_{t-1}|x_t)$ based on $q(x_{t-1}|x_t, x_0, x_{res}, x_{in})$.

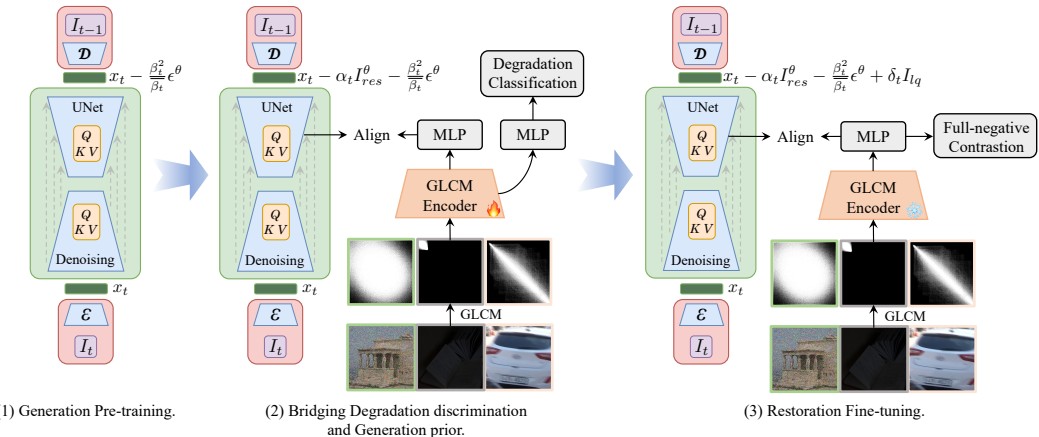

Figure 2: Three training stages in BDG. (1) During the generation stage, the model focuses on obtaining generation priors. (2) In the bridging stage, the MAS-GLCM, which can identify degradation fine-grainedly, is aligned with the features of the pre-trained generation model, thereby endowing the model with initial capabilities in degradation discrimination. (3) In the restoration stage, the model is tasked with performing restoration.

$$\mathcal{L}_{gen} = D_{KL}(q(x_{t-1}|x_t, x_0, x_{res}, x_{in})||p_\theta(x_{t-1}|x_t))$$
$$= \mathbb{E}_{q(x_t|x_0)} \left[ \|\mu(x_t, x_0) - \mu_\theta(x_t, t)\|^2 \right] \quad (5)$$
$$= \mathbb{E}_{t, \epsilon, x_{res}} \left[ \|\alpha_t(x_{res}^\theta - x_{res}) + \frac{\beta_t^2}{\bar{\beta}_t}(\epsilon^\theta - \epsilon)\|^2 \right].$$

Next, to effectively integrate the fine-grained degradation classification capability with the generation process, we introduce a novel degradation-generation bridging strategy. Specifically, a neural network is utilized to extract the abstract features of MAS-GLCM $M_{mas}$. Given that $M_{mas}$ demonstrates a robust fine-grained degradation classification capability, its high-dimensional features $F_{mas}$ are considered to embody this capability. Furthermore, these features are aligned with the intermediate features of the Diffusion Model $F_{diff}$. [1] The loss function is as follows.

$$\mathcal{L}_{bridge} = \frac{1}{2}\mathbb{E}[\mathrm{H}(y^{\mathrm{m2d}}(F_{mas}), p^{\mathrm{m2d}}(F_{mas})) + \\ \mathrm{H}(y^{\mathrm{d2m}}(F_{diff}), p^{\mathrm{d2m}}(F_{diff}))], \quad (6)$$

where $p^{\mathrm{m2d}}(F_{mas})$ is the soft-maxed similarity between $F_{mas}$ and $F_{diff}$, $y^{\mathrm{m2d}}$ denotes the one-hot ground-truth similarity of $F_{mas}$, and H is the cross-entropy function.

Finally, it is imperative to employ a loss function to ensure that $F_{mas}$ has fine-grained degradation classification capabilities. As illustrated in Figure 2 (2), an additional MLP is used to process $F_{mas}$, which is then optimized using the loss of degradation classification.

$$\mathcal{L}_{deg\text{-}cls} = \mathrm{H}(\mathrm{MLP}(F_{mas}), C), \quad (7)$$

where $C$ is the one-hot degradation class.

In **real-world** scenarios, degradation is challenging to categorize into distinct classes. A sound approach to emulate real-world degradation involves the fusion of simple degradations in multiple orders (Wang et al.; Zhang et al., 2021). For example, Real-ESRGAN (Wang et al.) exemplifies this by accumulating four types of degradation: blur, downsampling, JPEG compression, sinc artifacts and noise, iteratively superimposed eight steps. Each step in this chain represents an increased

---

[1]We select the first-layer feature of the UNet decoder as $F_{diff}$.

level of degradation complexity. Accordingly, we define eight intermediate states (e.g., after the first operation, second, etc.), which serve as pseudo-labels indicating the stage (or "order") of degradation application. These orders act as surrogates for degradation severity and compositional complexity. By training the model to recognize these order levels, it learns to implicitly estimate how heavily an image has been degraded, enabling better adaptation to varying degrees of real-world distortion. This provides a more feasible and meaningful learning signal than attempting to assign discrete type labels that may not reflect actual conditions for real-world super-resolution task. This task is termed "order classification", and its associated loss function can be derived by replacing $C$ in Eq. 7 with the one-hot order class.

In summary, the loss function at this stage is as follows.

$$\mathcal{L}_{bdg} = L_{gen} + \lambda(\mathcal{L}_{bridge} + \mathcal{L}_{deg\text{-}cls}), \tag{8}$$

where $\lambda$ balances these losses and is set to 0.1 by default.

**Restoration Fine-tuning.** Subsequent to the bridging stage, it is imperative to enhance the fidelity of predicted images and the fine-grained degradation discrimination ability. Specifically, it is crucial to ensure that the $x^{\theta}_{gt}$ predicted by the diffusion model aligns closely with the ground truth $x_{gt}$ while allowing the features of the diffusion model to accurately discern degradation using $\mathcal{L}_{bridge}$. Thus, we define $\mathcal{L}_{rft}$ as follows:

$$\mathcal{L}_{rft} = ||x^{\theta}_{gt} - x_{gt}||_1 + \lambda\mathcal{L}_{bridge}. \tag{9}$$

In **real-world** scenarios, the degradation observed in images is distinctly different. To improve the model's ability to recognize degradation, we reframe the degradation classification problem during the bridging stage as a contrastive learning task that involves only negative samples ("full negative contrastive learning"). Negative samples are those exhibiting different types or levels of degradation, and our goal is to extend the distance between pretrained MAS-GLCM's features $F_{mas}$ for each negative sample.

$$\mathcal{L}_{fcnl} = \sum_{i \in \mathcal{B}_1} \sum_{j \in \mathcal{B}_2} (1 - \cos(F^i_{mas}, F^j_{mas})), \tag{10}$$

where $\cos(f^i, f^j)$ denotes the cosine similarity between vectors $f^i$ and $f^j$. Within real-world super resolution task, $\mathcal{L}_{rft} = ||x^{\theta}_{gt} - x_{gt}||_1 + \lambda(\mathcal{L}_{bridge} + \mathcal{L}_{fcnl})$

It is important to note that we do not implement this loss during the bridging stage. In the bridging stage, the feature extractor of $M_{mas}$ is still training, and full negative contrastive learning would result in a representation collapse (Hu et al., 2021). In contrast, in the RFT stage, the weights of the feature extractor are frozen, rendering the model immune to the effects of representation collapse.

## 4 EXPERIMENTS

We perform an evaluation of our BDG across three distinct restoration tasks. **(1) All-in-one**: a model is trained to restore images in multiple degradation, including *real-world* scenarios. **(2) Mixed degradation**: a model restores images affected by composite degradation. **(3) Real-world**: real-world super-resolution task is also used to test BDG.

In the all-in-one and mixed degradation image restoration task, we employ a 36M UNet pre-trained on ImageNet. In the real-world super-resolution task, Stable Diffusion 2 (Rombach et al., 2022) is utilized as a baseline without incorporating additional architectures such as cross-attention or control-net. The implementation details are: batch size 256, learning rate $3 \times 10^{-4}$, and AdamW optimizer with $(\beta_1, \beta_2) = (0.9, 0.95)$. For each task, we train 300k iterations. The bridging stage and the restoration fine-tuning stage each have 150k iterations.

Detailed datasets, metrics, and qualitative results are provided in Appendix C.

## 4.1 ALL-IN-ONE IMAGE RESTORATION

We train a 5D all-in-one image restoration model with simulated dataset following DiffUIR (Zheng et al., 2024). This model is validated on simulated and real-world scenarios.

| Method | Deraining (6sets) | | Enhancement | | Desnowing | | Dehazing | | Deblurring (4sets) | |
|---|---|---|---|---|---|---|---|---|---|---|
| | PSNR ↑ | SSIM ↑ | PSNR ↑ | SSIM ↑ | PSNR ↑ | SSIM ↑ | PSNR ↑ | SSIM ↑ | PSNR ↑ | SSIM ↑ |
| Prompt-IR | 29.56 | 0.888 | 22.89 | 0.847 | 31.98 | 0.924 | 32.02 | 0.952 | 27.21 | 0.817 |
| DA-CLIP | 28.96 | 0.853 | 24.17 | 0.882 | 30.80 | 0.888 | 31.39 | 0.983 | 25.39 | 0.805 |
| DiffUIR-L | 31.03 | 0.904 | 25.12 | 0.907 | 32.65 | 0.927 | 32.94 | 0.956 | 29.17 | 0.864 |
| InsturctIR† | 31.35 | 0.911 | 24.33 | 0.887 | 32.71 | 0.934 | 32.08 | 0.957 | 29.58 | 0.874 |
| RAM-PromptIR† | 32.17 | 0.914 | 24.88 | 0.891 | 32.75 | 0.939 | 33.79 | 0.976 | 29.76 | 0.871 |
| DCPT-PromptIR† | 32.29 | 0.921 | 25.39 | 0.893 | 32.79 | 0.941 | 32.94 | 0.956 | 30.32 | 0.888 |
| **BDG (Ours)** | **34.75** | **0.974** | **27.42** | **0.930** | **32.86** | **0.950** | **34.33** | **0.993** | **31.11** | **0.904** |

Table 2: *All-in-one Image Restoration* results. † means the methods are retrained within datasets we used for fair comparison. The best and second results are shown in red and blue respectively.

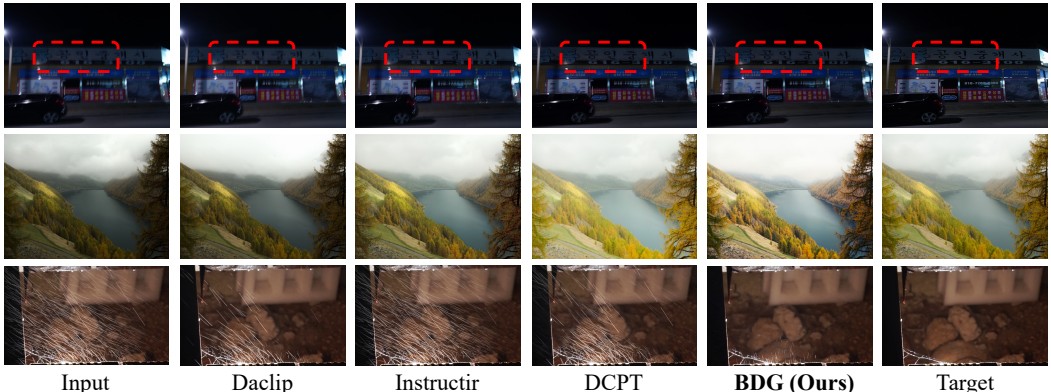

| Input | Daclip | Instructir | DCPT | **BDG (Ours)** | Target |

Figure 3: *Visual comparison on the 5D all-in-one image restoration task.* From top to bottom, each row corresponds to: deblurring, low-light enhancement, and deraining.

**Results** in 5D all-in-one task are reported in Table 2. Our BDG attains the State-of-The-Art (SoTA) performance across all tasks. Compared to DiffUIR (Zheng et al., 2024), which employs the same architecture and a similar diffusion sampling process, significant improvements are realized, measuring 3.72 dB, 2.30 dB, 1.39 dB, and 1.94 dB for deraining, low light enhancement, dehazing, and deblurring, respectively. In contrast to the recently leading method DCPT (Hu et al., 2025a), an enhancement of 2.46 dB is also observed in deraining. In particular, the restoration fidelity of the large-scale unified visual generation model (Wang et al., 2023b) is inferior to that of the model specifically trained for restoration.

| Degradation | Snow | Haze | Low-light |
|---|---|---|---|
| Method ↓ | PIQE ↓ / BRISQUE ↓ | PIQE ↓ / BRISQUE ↓ | PIQE ↓ / BRISQUE ↓ |
| DA-CLIP | 31.34 / 24.45 | 47.67 / 34.90 | 37.64 / 27.45 |
| InstructIR | 33.35 / 24.41 | 50.97 / 31.45 | 36.08 / 26.31 |
| DCPT-NAFNet | 32.59 / 25.02 | 52.40 / 37.97 | 35.48 / 26.97 |
| UniRestore | 32.69 / 27.16 | 46.88 / 30.95 | 34.63 / 27.05 |
| FoundIR | 33.18 / 26.20 | 61.14 / 42.26 | 44.17 / 33.51 |
| **BDG (Ours)** | **31.45 / 24.00** | **47.59 / 34.75** | **34.44 / 27.41** |

Table 3: *Real-world restoration results* in four real-world degradation types under the zero-shot setting. The best and second results are shown in red and blue respectively.

**Results in real-world scenarios** are reported in Table 3. According to these quantitative metrics, BDG attains the majority of the best and the second-best results, notably achieving the lowest PIQE in the low-light enhancement task. In other real-world degradations, BDG also achieves the lowest or the second-lowest PIQE and BRISQE, demonstrating its robustness. In comparison to diffusion-based methods lacking degradation identification (Zheng et al., 2024; Li et al., 2025), BDG demonstrates enhanced fidelity while maintaining the detailed texture restoration prowess inherent to the generation model.

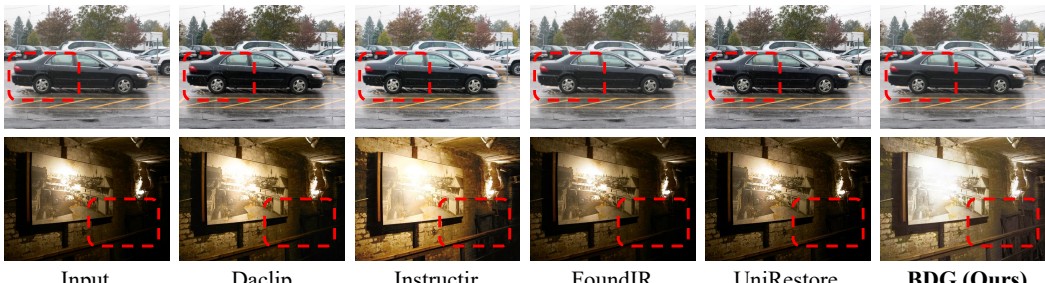

| Input | Daclip | Instructir | FoundIR | UniRestore | **BDG (Ours)** |

Figure 4: *Visual comparison on the real-world all-in-one image restoration task.* From top to bottom, each row corresponds to: desnowing and low-light enhancement.

## 4.2 IMAGE RESTORATION UNDER MIXED DEGRADATION

We also train and test our BDG for the mixed degradation scenarios (Guo et al., 2025).

| Method | CDD11-Double | | | | | | | | | | | CDD11-Triple | | | |
| --- | --- | --- | --- | --- | --- | --- | --- | --- | --- | --- | --- | --- | --- | --- | --- |
| | L+H | | L+R | | L+S | | H+R | | H+S | | | L+H+R | | L+H+S | |
| PromptIR | 24.49 | .789 | 25.05 | .771 | 24.51 | .761 | 24.54 | .924 | 23.70 | .925 | | 23.74 | .752 | 23.33 | .747 |
| WGWSNet | 24.27 | .800 | 25.06 | .772 | 24.60 | .765 | 27.23 | .955 | 27.65 | .960 | | 23.90 | .772 | 23.97 | .771 |
| WeatherDiff | 21.83 | .756 | 22.69 | .730 | 22.12 | .707 | 21.25 | .868 | 21.99 | .868 | | 21.23 | .716 | 21.04 | .698 |
| OneRestore | 25.79 | .822 | 25.58 | .799 | 25.19 | .789 | 29.99 | .957 | 30.21 | .964 | | 24.78 | .788 | 24.90 | .791 |
| MoCE-IR | 26.24 | .817 | 26.25 | .800 | 26.04 | .793 | 29.93 | .964 | 30.19 | .970 | | 25.41 | .789 | 25.39 | .790 |
| **BDG (Ours)** | **27.27** | **.833** | **26.67** | **.817** | **26.59** | **.809** | **34.21** | **.975** | **34.42** | **.979** | | **26.14** | **.809** | **26.45** | **.809** |

Table 4: *Comparison to state-of-the-art on composited degradations.* The best and second results are shown in red and blue respectively.

**Results** in mixed degradation scenarios are reported in Table 4. Compared to the previous SoTA method (Zamfir et al., 2025), BDG demonstrates substantial performance improvements across all mixed degradation scenarios, with particularly notable improvements in scenarios characterized by haze and rain (H+R) degradations, where the enhancement reaches 4.28 dB.

## 4.3 REAL-WORLD SUPER-RESOLUTION

We conduct experiments on real-world super-resolution.

| Datasets | Metrics | BSRGAN | Real-ESRGAN | FeMaSR | StableSR | SUPIR | SeeSR | DiffBIR | PASD | LDM | ResShift | **BDG (Ours)** |
| --- | --- | --- | --- | --- | --- | --- | --- | --- | --- | --- | --- | --- |
| *DIV2K-Val* | PSNR ↑ | 21.87 | 21.94 | 20.85 | 20.84 | 18.68 | 21.19 | 20.94 | 20.77 | 21.26 | 21.75 | **24.1977** |
| | SSIM ↑ | 0.5539 | 0.5736 | 0.5163 | 0.4887 | 0.4664 | 0.5386 | 0.4938 | 0.4958 | 0.5239 | 0.5422 | **0.6241** |
| | LPIPS ↓ | 0.4136 | 0.3868 | 0.3973 | 0.4055 | 0.4102 | 0.3843 | 0.4270 | 0.4410 | 0.4154 | 0.4284 | **0.3669** |
| | DISTS ↓ | 0.2737 | 0.2601 | 0.2428 | 0.2542 | 0.2207 | 0.2257 | 0.2471 | 0.2538 | 0.2500 | 0.2606 | **0.2571** |
| | FID ↓ | 64.28 | 53.46 | 53.7 | 36.57 | 32.18 | 31.93 | 40.42 | 40.77 | 41.93 | 55.77 | **43.49** |
| | MANIQA ↑ | 0.4834 | 0.5251 | 0.4869 | 0.5914 | 0.5491 | 0.6198 | 0.6205 | 0.6049 | 0.5237 | 0.5232 | **0.5066** |
| | MUSIQ ↑ | 59.11 | 58.64 | 58.1 | 62.95 | 65.33 | 68.33 | 65.23 | 66.85 | 56.52 | 58.23 | **61.2826** |
| | CLIPIQA ↑ | 0.5183 | 0.5424 | 0.5597 | 0.6486 | 0.6035 | 0.6946 | 0.6664 | 0.6799 | 0.5695 | 0.5948 | **0.6396** |
| *DrealSR* | PSNR ↑ | 28.75 | 28.64 | 26.9 | 28.13 | 24.41 | 28.17 | 26.76 | 27 | 27.98 | 28.46 | **28.7961** |
| | SSIM ↑ | 0.8031 | 0.8053 | 0.7572 | 0.7542 | 0.6696 | 0.7691 | 0.6576 | 0.7453 | 0.7673 | | **0.8039** |
| | LPIPS ↓ | 0.2883 | 0.2847 | 0.3169 | 0.3315 | 0.3844 | 0.3189 | 0.4599 | 0.3931 | 0.3405 | 0.4006 | **0.3282** |
| | DISTS ↓ | 0.2142 | 0.2089 | 0.2235 | 0.2263 | 0.2448 | 0.2315 | 0.2749 | 0.2515 | 0.2259 | 0.2656 | **0.2774** |
| | MANIQA ↑ | 0.4878 | 0.4907 | 0.442 | 0.5591 | 0.457 | 0.6042 | 0.5923 | 0.5043 | 0.4586 | | **0.4899** |
| | MUSIQ ↑ | 57.14 | 54.18 | 53.74 | 58.42 | 64.53 | 64.93 | 61.19 | 64.81 | 53.73 | 50.6 | **58.7432** |
| | CLIPIQA ↑ | 0.4915 | 0.4422 | 0.5464 | 0.6206 | 0.58 | 0.6804 | 0.6346 | 0.6773 | 0.5706 | 0.5342 | **0.6053** |
| *RealSR* | PSNR ↑ | 26.39 | 25.69 | 25.07 | 24.7021 | 22.67 | 25.18 | 24.77 | 24.29 | 25.48 | 26.31 | **25.5105** |
| | SSIM ↑ | 0.7654 | 0.7616 | 0.7358 | 0.7085 | 0.6567 | 0.7216 | 0.6572 | 0.663 | 0.7148 | 0.7421 | **0.7509** |
| | LPIPS ↑ | 0.267 | 0.2727 | 0.2942 | 0.3002 | 0.3545 | 0.3019 | 0.3658 | 0.3435 | 0.318 | 0.346 | **0.3016** |
| | DISTS ↓ | 0.2121 | 0.2063 | 0.2288 | 0.2139 | 0.2385 | 0.2223 | 0.231 | 0.2259 | 0.2213 | 0.2498 | **0.2574** |
| | MANIQA ↑ | 0.5399 | 0.5487 | 0.4865 | 0.6221 | 0.5396 | 0.6442 | 0.6253 | 0.6493 | 0.5423 | 0.5285 | **0.5578** |
| | MUSIQ ↑ | 63.21 | 60.18 | 58.95 | 65.78 | 66.09 | 69.77 | 64.85 | 68.69 | 58.81 | 58.43 | **64.6183** |
| | CLIPIQA ↑ | 0.5001 | 0.4449 | 0.527 | 0.6178 | 0.5171 | 0.6612 | 0.6386 | 0.659 | 0.5709 | 0.5444 | **0.6332** |

Table 5: *Real-world super resolution results* on synthetic and real-world benchmarks. The best and second best results of each metric in diffusion-based methods are highlighted in red and blue, respectively.

**Results** in real-world super-resolution are shown in Table 5. We have the following observations. (1) Our BDG consistently scores the highest or second highest in PSNR, SSIM, and LPIPS across all datasets. (2) BDG shows a notable improvement in fidelity. In DIV2K-Val, BDG outperforms 2.45 dB in comparison to the second-best diffusion method and all GAN-based methods in PSNR. This huge enhancement is because diffusion-based methods often generate textures that deviate from the ground truth, putting the results at a disadvantage in full-reference metrics. In contrast, BDG closely aligns the output with LQ images through Eq. 4, securing favorable results in full-reference metrics. (3) In non-reference metrics such as MANIQA, MUSIQ and CLIPIQA, BDG outperforms its baseline (Rombach et al., 2022) and ResShift (Yue et al., 2023). BDG effectively informs the model about the type or level of degradation, avoiding the model from creating textures that are inconsistent with GT at lower levels of degradation. Overall, BDG excels in full-reference metrics while being competitive in non-reference metrics.

## 4.4 ABLATION STUDY

The results of the aforementioned experiments generally prove that BDG leads to a significant gain in restoration performance. In this subsection, we mainly analyze the impact of different components in BDG on the restoration results. We perform ablation studies on 5D all-in-one image restoration and real-world super-resolution.

| Bridging | RFT | PSNR / SSIM |
|---|---|---|
| 300k | 0 | 30.25 / 0.871 |
| 0 | 300k | 31.03 / 0.908 |
| 150k | 150k | **32.09 / 0.950** |

| Bridging | RFT | PSNR / SSIM / CLIPIQA |
|---|---|---|
| 300k | 0 | 28.35 / 0.7988 / 0.5839 |
| 0 | 300k | 28.73 / 0.8093 / 0.4787 |
| 150k | 150k | **28.80 / 0.8039 / 0.6053** |

Table 6: Ablation of training stages on all-in-one restoration task (left) and real-world super-resolution task (right).

**Impact of training stages.** As shown in Table 6, the optimal restoration performance is achieved when both the bridging stage and the RFT stage are present.

| $\mathcal{L}_{gen}$ | $\mathcal{L}_{bridge}$ | $\mathcal{L}_{deg\text{-}cls}$ | PSNR / SSIM |
|---|---|---|---|
| ✔ | ✘ | ✘ | 31.11 / 0.883 |
| ✔ | ✔ | ✘ | 20.88 / 0.847 |
| ✔ | ✔ | ✔ | **32.09 / 0.950** |

| $\mathcal{L}_{bridge}$ | $\mathcal{L}_{fcnl}$ | PSNR / SSIM / CLIPIQA |
|---|---|---|
| ✘ | ✘ | 27.57 / 0.7821 / 0.5839 |
| ✘ | ✔ | 28.23 / 0.7988 / 0.5935 |
| ✔ | ✘ | 27.88 / 0.7844 / 0.5589 |
| ✔ | ✔ | **28.80 / 0.8039 / 0.6053** |

Table 7: Ablation of losses in the bridging stage (left) and the RFT stage (right).

**Impact of losses.** As demonstrated in Table 7, the loss functions that we have developed for the bridging stage and the RFT stage result in improvements in restoration fidelity while maintaining perceptual integrity. The performance of the model deteriorates markedly when reliance is not placed on $L_{deg\text{-}cls}$. We contend that in the absence of $L_{deg\text{-}cls}$, the MAS-GLCM encoder is devoid of discrimination objectives, leading to a model collapse issue. Consequently, the diffusion features become aligned with the collapsed MAS-GLCM encoder, resulting in suboptimal results. In contrast, with only $L_{gen}$, the collapsed MAS-GLCM encoder does not adversely impact the restoration models, thus still achieving a certain degree of restoration performance.

## 5 CONCLUSION

This paper presents Bridging Degradation discrimination and Generation (BDG) for universal image restoration. The BDG approach proficiently enhances the model's capability to perform restoration contingent upon type or level of input degradation while effectively leveraging the generative prior to enrich the detail and texture of the output image. Specifically, we design MAS-GLCM to finely identify the degradation. Subsequently, by reformulating the diffusion backward process equation, we design a three-stage diffusion training method. It endows the model with the ability to discern degradation while preserving its capacity to generate superior texture details by aligning the MAS-GLCM with the diffusion features. We substantiate the efficacy of BDG in all-in-one image restoration, mixed degradation image restoration and real-world super-resolution.

**Ethics Statement.** This paper presents work whose goal is to advance the field of image restoration. There are many potential societal consequences of our work. Given the increasing capabilities of image restoration techniques, we advocate avoiding the misuse of related technologies, such as forging misleading images or restoring and enhancing images for malicious purposes.

**Reproducibility Statement.** We state that BDG is highly reproducible. Implementation and datasets details and on our main experiences are provided in Section 4 and Appendix C. It is anticipated that these descriptions can sufficiently demonstrate the reproducibility of BDG. We plan to open-source the code and weight files after the paper passes peer review.

## ACKNOWLEDGEMENT

This work was supported in part by National Key Research and Development Program of China (2025YFA1805700); in part by National Natural Science Foundation of China (82371112, 62501020); in part by the Science Foundation of Peking University Cancer Hospital (JC202505).

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

## A  EXPERIMENTAL SETTINGS ON STUDIES ABOUT MAS-GLCM

**TSNE settings in Sec. 3.1.** In the TSNE visualization results, we select three weather-related degradations: haze, snow, and low-light, as well as two types of noise: Gaussian noise and Pepper noise. Degraded image data for hazy, low-light, and snowy conditions are obtained from SOTS, LOL, and Snow100K, respectively, while Gaussian and Pepper noise data are synthesized from the Kodak dataset with a Gaussian noise level of 25 and a Pepper noise ratio of 0.1. This experiment utilizes the TSNE function from "sklearn.manifold" to reduce the data dimensionality to a two-dimensional space, with the number of iterations set to 2k and computation accelerated using four CPU cores.

**Datasets used in Sec. 3.1.** We select the first 100 images from SOTS, LOL, and Snow100K as representative degraded image data for hazy, low-light, and snowy conditions, respectively, while the data with different noise types and noise levels are generated from the same Kodak dataset. Each image is center-cropped to 256 × 256 to ensure a consistent resolution. We use the default setting of the KNeighborsClassifier in the Python library sklearn, which assigns equal weights to all neighboring points and employs the Minkowski distance with the default parameter p=2, corresponding to the Euclidean distance.

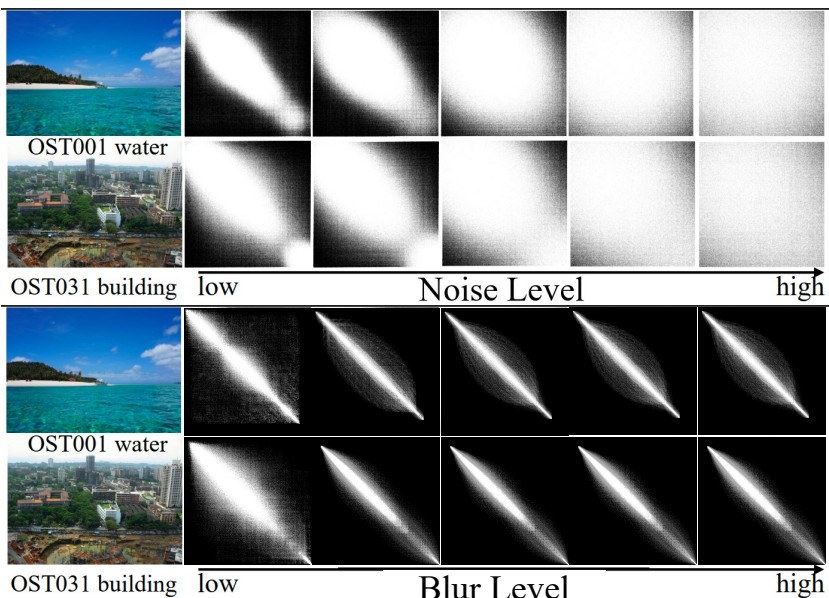

Figure 5: MAS-GLCM on images with different texture.

**More results on MAS-GLCM.  (1) MAS-GLCM robustness.** We argue that MAS-GLCM in different semantics is not significantly different, as shown in Figure 6. In the OST data set with seven different semantics, MAS-GLCM classifies the degradation well (77% in the noise level classification). **2) Real datasets.** We evaluate MAS-GLCM on real datasets: low-light (LOLv2), snow (Snow100k), haze (RTTS) and noise (SIDD). The result can be shown in Figure 6 (c). **3) Scaling datasets.** We also test MAS-GLCM on 5000 synthetic or real data, with the results shown in Figure 6. **4) Mixed datasets.** We conduct experiments on the CDD dataset, which contains six composite degradations: haze+rain, low+haze, low+rain, low+snow, low+haze+rain, and low+haze+snow. T-SNE visualizations of the clustering behavior in Figure 7. These show that MAS-GLCM successfully separates certain categories such as low+snow and low+rain from others. However, it struggles to fully distinguish more similar composite degradations, such as low+haze+rain and low+haze.

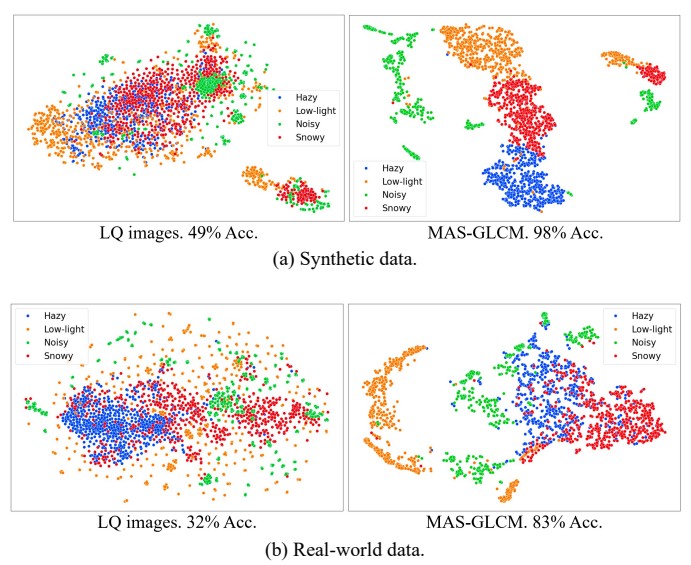

(a) Synthetic data.

(b) Real-world data.

Figure 6: More results on MAS-GLCM.

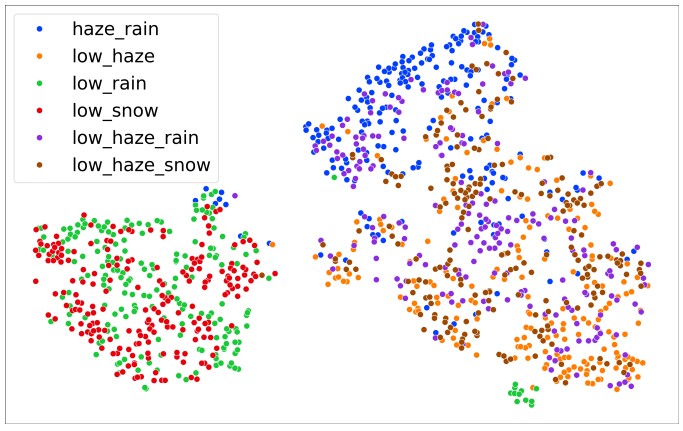

Figure 7: T-SNE results on mixed degradations.

## B  PROOF OF EQ. 4

The forward process of the diffusion model we use is as follows.

$$x_t = x_{t-1} + \alpha_t x_{res} + \beta_t \epsilon_{t-1} - \delta_t x_{lq}, \tag{11}$$

where $x_t$ is the diffusing result in timestep $t$, $x_{res} := x_{lq} - x_{hq}$ is the residual of the LQ image $x_{lq}$ and the HQ image $x_{hq}$. $\alpha_t$, $\beta_t$, and $\delta_t$ is the noise coefficient of $x_{res}$, standard Gaussian noise $\epsilon$, and $x_{lq}$, respectively.

Since we need to fit the distribution of $x_{hq}$, we set $x_0 = x_{hq}$. According to the Markov Chain, Eq. 3 can be reformulated as follows.

$$x_t = x_0 + \overline{\alpha_t} x_{res} + \overline{\beta_t} \epsilon - \overline{\delta_t} x_{lq}, \tag{12}$$

where $\overline{\alpha_t} = \sum_{i=1}^{t} \alpha_i$, $\overline{\beta_t} = \sqrt{\sum_{i=1}^{t} \beta_i^2}$, and $\overline{\delta_t} = \sum_{i=1}^{t} \delta_i$.

Once this diffusion model is trained, we simulate the distribution $p_\theta^t(x_{t-1}|x_t)$ through $q(x_{t-1}|x_t, x_{lq}, x_0^\theta, I_{res}^\theta)$, where $I_{res}^\theta$ is the predicted residual and $x_0^\theta = x_{lq} - x_{res}^\theta$ according to the definition of $x_{res}$.

Based on the Bayes' theorem, we can obtain the following.

$$
\begin{aligned}
p_\theta(x_{t-1}|x_t) &\rightarrow q(x_{t-1}|x_t, x_{in}, I_0^\theta, x_{res}^\theta) \\
&= q(x_t|x_{t-1}, x_{in}, x_{res}^\theta) \frac{q(x_{t-1}|I_0^\theta, x_{res}^\theta, x_{in})}{q(x_t|I_0^\theta, x_{res}^\theta, x_{in})} \\
&\propto exp\left[ -\frac{1}{2} \left( \left( \frac{\overline{\beta}_t^2}{\beta_t^2 \overline{\beta}_{t-1}^2} \right) x_{t-1}^2 - 2 \left( \frac{x_t + \delta_t x_{in} - \alpha_t x_{res}^\theta}{\beta_t^2} \right. \right. \right. \\
&\quad \left. \left. \left. + \frac{I_0^\theta + \overline{\alpha}_{t-1} x_{res}^\theta - \overline{\delta}_{t-1} x_{in}}{\overline{\beta}_{t-1}^2} \right) x_{t-1} + C(x_t, I_0^\theta, x_{res}^\theta, x_{in}) \right) \right].
\end{aligned}
\tag{13}
$$

As the goal of the formulation is to obtain the distribution of $x_{t-1}$, we simplify and rearrange it into a form about $x_{t-1}$ and $C(x_t, I_0^\theta, x_{res}^\theta, x_{in})$ is the term unrelated to it. So, the mean $\mu_\theta(x_t, t)$ and the variance $\sigma_\theta(x_t, t)$ of the distribution $p_\theta^t(x_{t-1}|x_t)$ are as follows.

$$
\begin{aligned}
\mu_\theta(x_t, t) &= x_t - \alpha_t I_{res}^\theta - \frac{\beta_t^2}{\overline{\beta}_t} \epsilon^\theta + \delta_t x_{lq}; \\
\sigma_\theta(x_t, t) &= \frac{\beta_t^2 \overline{\beta}_{t-1}^2}{\overline{\beta}_t^2}.
\end{aligned}
\tag{14}
$$

## C  EXPERIMENTS SETUP AND QUALITATIVE COMPARISONS

### C.1  ABLATION ON ANGLES AND SCALES IN MAS-GLCM

We evaluate several configurations with reduced or asymmetric angle and scale settings:

- Incomplete angles. For instance, using only non-negative angles (e.g., [0, 45, 90, 135, 180]) restricts the GLCM to capture co-occurrence patterns from directions above the current pixel, potentially missing symmetric or opposing texture structures.

- Reduced angles. Using a sparse set such as [-180, -90, 0, 90, 180] limits the model to only horizontal and vertical relationships, ignoring diagonal textures that are common in natural images.
- Limited scales. Reducing the number of distances decreases sensitivity to both fine-grained and coarse-level texture variations.

| angles | scales | Avg. PSNR (dB) / SSIM |
|---|---|---|
| $[0, 45, 90, 135, 180]$ | $[1, 3, 5]$ | 31.77 / 0.932 |
| $[-180, -90, 0, 90, 180]$ | $[-5, -1, 1, 5]$ | 31.93 / 0.944 |
| $[-180, -135, -90, -45, 0, 45, 90, 135, 180]$ | $[-5, -3, -1, 1, 3, 5]$ | 32.09 / 0.950 |

Table 8: Ablation on angles and scales in MAS-GLCM.

As shown in Table 8, when the selected angles and scales provide comprehensive spatial coverage, restoration performance is consistently strong. The full configuration achieves the best results, indicating that complete directional and scale diversity helps the model better characterize complex degradation patterns. However, performance degrades noticeably when critical directions or scales are omitted, particularly when symmetry or diagonal structures are neglected.

### C.2 5D ALL-IN-ONE RESTORATION

**Datasets.** For all-in-one image restoration, datasets are: Rain13k (Yang et al., 2017) and SynRain-13k (Li et al., 2022b), which contains 13,712 training images for deraining; LOL (Wei et al., 2018), which contains 485 training images and 15 test images for low-light enhancement; Snow100K Liu et al. (2018), which contains 50,000 training data, 50,000 testing data for desnowing; RESIDE (Li et al., 2018), which contains 72,135 training images and 500 test images (SOTS) for dehazing; GoPro (Nah et al., 2017) and RealBlur for motion deblurring. Following Zheng et al. (2024), we use the PSNR and SSIM calculated in the Y channel in the YCbCr space as metrics.

### C.3 MIXED DEGRADTAION

**Datasets.** For the mixed degradation restoration task, we use the CDD (Guo et al., 2025), which consists of 11 degradations (rain, low light, snow, and their and combinations). It has 13,013 image pairs for training and 2,200 for testing. Following Hu et al. (2025a), we use the PSNR and SSIM calculated in the sRGB space as metrics.

**Qualitative comparisons.**

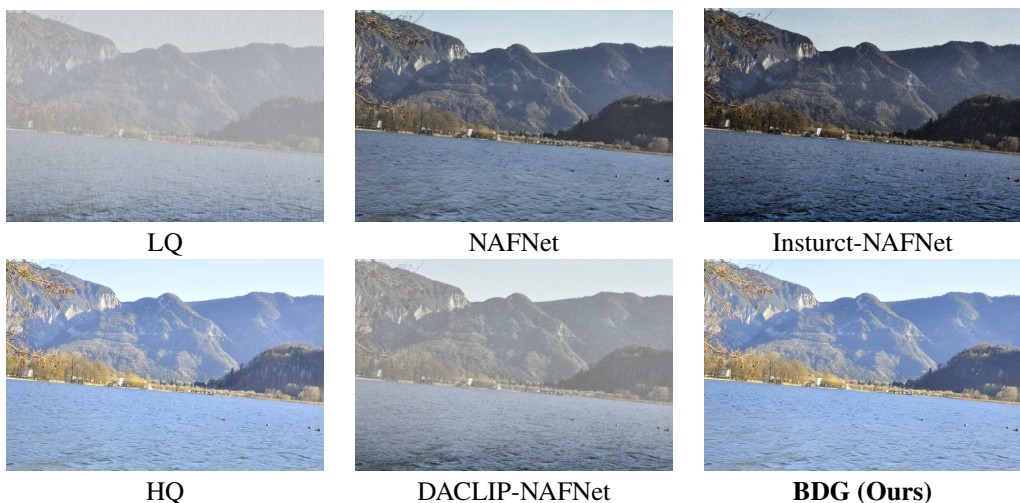

Figure 8: *Visual comparison on low-light + haze + rain.*

## C.4 REAL-WORLD SUPER-RESOLUTION

**Datasets.** Following Wang et al. (2024); Wu et al. (2024), we use the LSDIR (Li et al., 2023) and the first 10k images of FFHQ (Karras et al., 2019) for training data. Training pairs are synthesized via Real-ESRGAN (Wang et al.). For evaluation, we employ the following test sets: (1) We extract 3k randomly cropped $512 \times 512$ resolution patches from the DIV2K validation set Agustsson & Timofte (2017), which are subsequently degraded using the same pipeline as used during training. This dataset is henceforth referred to as DIV2K-Val. (2) Additionally, we use center-cropped RealSR (Cai et al., 2019) and DRealSR (Wei et al., 2020) as real-world benchmarks, following Wang et al. (2024).

**Metrics.** To offer a comprehensive assessment of the performance of the various methods, we engage in a spectrum of reference and non-reference metrics. PSNR and SSIM, computed in the Y channel in the YCbCr color space, are used as reference-based fidelity metrics, while LPIPS (Zhang et al., 2018) and DISTS (Ding et al., 2020) serve as reference-based perceptual quality metrics. The FID statistic (Heusel et al., 2017) assesses the distributional divergence between the original and reconstructed images. In addition, MANIQA (Yang et al., 2022), MUSIQ (Ke et al., 2021), and CLIPIQA (Wang et al., 2023a) are implemented as non-reference image quality metrics.

**Qualitative comparisons.**

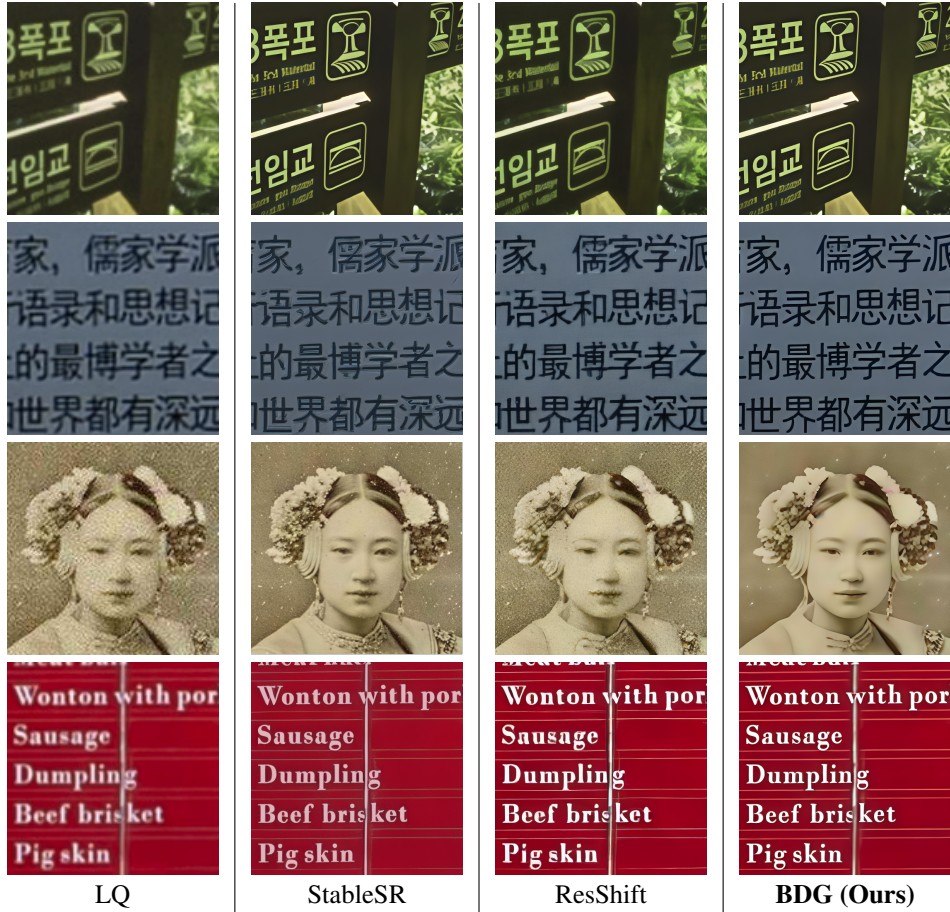

| LQ | StableSR | ResShift | **BDG (Ours)** |

Figure 9: BDG handles complex noise and text details in images well, but still faces certain over-smoothing problems. Please zoom in for better view.

## D   LIMITATIONS AND FUTURE WORK

There remains considerable scope for advancing the design of MAS-GLCM and BDG. Specifically, the current implementation of MAS-GLCM does not accommodate the detection of color deviations

or global geometric transformations, and it may exhibit sensitivity to image resolution. Moreover, BDG depends on a relatively complex three-stage training paradigm. In future work, we intend to extend the generalization capability of MAS-GLCM to a broader range of low-level vision tasks and to further simplify and simplify the three-stage training strategy employed in BDG.

## E   THE USE OF LARGE LANGUAGE MODELS (LLMS)

LLMs are used to correct potential grammatical inaccuracies in the manuscript. LLMs do not participate in research ideation.

