# OpenReview forum: "Bridging Degradation Discrimination and Generation for Universal Image Restoration"
_ICLR.cc/2026/Conference — ICLR 2026 Poster_

### Official Review · Reviewer_JVFR · 2025-10-30

**Soundness:** 3
**Presentation:** 3
**Contribution:** 3
**Rating:** 6
**Confidence:** 3

**Summary:**

The method proposed in this paper follows a 'generate-then-restore' paradigm. Initially, a generative approach is applied to obtain a high-quality output. To compensate for potential deviations from the original image introduced during the generation process, two additional steps are employed. The first step uses degradation information to guide the generative model toward restoration, biasing the output toward the content of the input low-quality image; this degradation information is extracted using traditional operators, such as the gray-level co-occurrence matrix. The second step directly incorporates the low-quality image itself, further ensuring that the result follows the original content of the image.

**Strengths:**

The paper integrates fine-grained degradation information (MAS-GLCM features) with the diffusion generative prior, achieving both high-fidelity restoration and rich texture generation. Compared to traditional full-reference methods or purely generative approaches, it effectively balances fidelity and perceptual quality.

**Weaknesses:**

1. The paper provides only six visual results, which is insufficient. Although some quantitative improvements are reported, improvements in low-level metrics do not necessarily correspond to better perceptual quality. Therefore, the current results are not sufficient to fully demonstrate the effectiveness of the proposed method; more visual examples are needed.

2. MAS-GLCM is a hand-crafted feature, which may perform poorly on certain complex or previously unseen types of degradation. It may also be limited for non-grayscale or non-texture-related degradations, such as color shifts or geometric distortions.

3. The training procedure is relatively complex, requiring a total of three stages.

4. The three-stage training strategy combined with multiple loss functions adds additional complexity and requires careful hyperparameter tuning (e.g., the λ coefficient).

5. Although the bridging and restoration fine-tuning stages improve fidelity, the generated content may still slightly deviate from the original image under extreme or composite degradation scenarios. Moreover, the performance on extreme or previously unseen real-world degradations remains unclear.

**Questions:**

Please explain why there are not more visual results, and analyze the performance of the algorithm in real-world and diverse scenarios.

---

> ### Author Response · Authors · 2025-11-25
>
> Thank you for your insightful comments. We first list your advice and questions, give our detailed answers, and **argue that your concerns can be addressed**.
>
> > Weakness 1 & Question 1. The paper provides only six visual results, which is insufficient. Although some quantitative improvements are reported, improvements in low-level metrics do not necessarily correspond to better perceptual quality. Therefore, the current results are not sufficient to fully demonstrate the effectiveness of the proposed method; more visual examples are needed. Please explain why there are not more visual results, and analyze the performance of the algorithm in real-world and diverse scenarios.
>
> **A.** We fully agree with your comment. Due to time constraints, we did not include a sufficient number of visualization results in the original submission, and we sincerely apologize for this omission. In response, as the main paper has been extended to ten pages, we have also incorporated several representative visual examples into the main text to improve clarity and support our analysis.
>
> Specifically, we added:
> - **Figure 3** for 5D all-in-one image restoration, including deblurring, low-light enhancement, and deraining.
> - **Figure 4** for real-world image restoration, including desnowing and low-light enhancement.
>
> We believe these additions significantly strengthen the interpretability and completeness of the paper, and we appreciate your constructive feedback that helped us improve the presentation.

---

> ### Author Response · Authors · 2025-11-25
>
> > Weakness 2-1. MAS-GLCM is a hand-crafted feature, which may perform poorly on certain complex or previously unseen types of degradation.
>
> **A.** We would like to clarify that, as a non-learnable and handcrafted feature extraction method, MAS-GLCM does not rely on training data and trainable parameters. Therefore **does not have the concept of "unseen" degradations**. It operates directly on pixel intensity statistics and is inherently applicable to any input without requiring prior exposure to specific degradation types.
>
> However, we fully acknowledge that **MAS-GLCM may still face challenges in discriminating highly complex or heavily mixed degradations**, particularly when multiple distortions interact in non-additive ways. We appreciate the opportunity to discuss this limitation and provide a detailed analysis from multiple perspectives. Specifically, we evaluate the performance of MAS-GLCM in distinguishing three types of complex degradation scenarios.
>
> **(1/3) Different degradation levels.**
> We first clarify that MAS-GLCM demonstrates strong capability in distinguishing different levels of a single degradation type. As shown in our response to the Weakness 1 you raised, MAS-GLCM achieves superior performance in both degradation type and level classification compared to existing methods. This provides direct evidence that MAS-GLCM can effectively capture meaningful variations associated with specific degradation types and their severity.
>
> **(2/3) Mixed degradation.**
> We acknowledge that the ability of MAS-GLCM to discriminate among mixed degradation types is a little limited. To evaluate this quantitatively and qualitatively, we conduct experiments on the CDD dataset, which contains six composite degradations: `haze+rain`, `low+haze`, `low+rain`, `low+snow`, `low+haze+rain`, and `low+haze+snow`.
>
> - *Quantitative results.* We compare the classification accuracy of various degradation representations under these mixed conditions:
>
> | Method | Acc (%) |
> |--|--|
> | LQ images | 23.41 |
> | Sobel (grad) | 25.92 |
> | Laplace (grad) | 37.75 |
> | Fourier | 29.50 |
> | PromptIR | 16.67 |
> | DCPT | 16.67 |
> | Text with DA-CLIP | 16.67 |
> | MAS-GLCM | 54.42 |
>
> MAS-GLCM achieves the highest accuracy, indicating it still retains discriminative power even under mixed distortions. However, the accuracy of MAS-GLCM remains below 60%.
>
> - *Qualitative results.* We also provide T-SNE visualizations of the clustering behavior in Appendix A. These show that MAS-GLCM successfully separates certain categories such as `low+snow` and `low+rain` from others. However, it struggles to fully distinguish more similar composite degradations, such as `low+haze+rain` and `low+haze`.
>
> In summary, while MAS-GLCM is indeed constrained in its ability to disentangle mixed degradations, both quantitative and qualitative results confirm that its discrimination performance remains stronger than existing alternatives.
>
> **(3/3) Real-world scenarios.**
>
> In real-world scenarios, degradation types are diverse and heavily intertwined. As you rightly point out, the inability to precisely identify or decompose individual degradation components may limit the performance of our BDG.
>
> To address this limitation, we propose "order classification" as an alternative strategy as we stated in line 300-307 of the maintext. Instead of categorizing degradation by type, **"order classification" classifies samples based on the sequential order and cumulative complexity of degradation operations applied during the synthesis process.** For instance, in the degradation pipeline inspired by Real-ESRGAN, distortions are progressively introduced following the sequence:
> `+Blur -> +Resize -> +Noise -> +JPEG -> +Blur -> +Resize -> +Noise -> +JPEG`.
>
> Each step in this chain represents an increased level of degradation complexity. Accordingly, we define eight intermediate states (e.g., after the first operation, second, etc.), which serve as pseudo-labels indicating the stage (or "order") of degradation application. These orders act as surrogates for degradation severity and compositional complexity.
>
> By training the model to recognize these order levels, it learns to implicitly estimate how heavily an image has been degraded, enabling better adaptation to varying degrees of real-world distortion. This provides a more feasible and meaningful learning signal than attempting to assign discrete type labels that may not reflect actual conditions for real-world super-resolution task.
>
> **However**, we think that this is a pragmatic compromise rather than a complete solution. It does not enable fine-grained disentanglement of mixed degradations, nor does it allow precise estimation of mixture proportions. Addressing these limitations remains an open challenge, which we plan to explore in future work.

---

> ### Author Response · Authors · 2025-11-25
>
> > Weakness 2-2. It may also be limited for non-grayscale or non-texture-related degradations, such as color shifts or geometric distortions.
>
> **A.** We sincerely thank the reviewer for this insightful comment. You are correct that MAS-GLCM, as a gray-level co-occurrence-based descriptor, is primarily sensitive to intensity and texture statistics and may have limited sensitivity to certain types of degradations such as color shifts or geometric distortions (e.g., rotation).
>
> Let us clarify the scope and design intent:
>
> **(1/2) Color shifts.**
>
> While MAS-GLCM operates on luminance (Y channel in YCbCr) by default, it does not directly model chrominance changes such as tint, hue shift, or white balance errors. However, strong color imbalances often introduce unnatural pixel value distributions and local intensity anomalies at edges, which may indirectly affect the MAS-GLCM patterns.
>
> **(2/2) Geometric distortions.**
>
> Global geometric transformations (e.g., rotation) do not typically degrade image quality. Local geometric distortions that arise from lens aberrations or atmospheric turbulence may manifest as texture warping, which could partially influence GLCM responses if they alter spatial regularity. However, pure alignment errors are indeed outside the modeling capacity of MAS-GLCM.
>
> Based on above analyze, we fully agree that handling color and geometric factors requires specialized mechanisms. We have added a discussion of these limitations in the revised manuscript and will explore hybrid descriptors that integrate color and spatial transformation cues in future work.

---

> ### Author Response · Authors · 2025-11-25
>
> > Weakness 3 & Weakness 4. The training procedure is relatively complex, requiring a total of three stages. The three-stage training strategy combined with multiple loss functions adds additional complexity and requires careful hyperparameter tuning (e.g., the λ coefficient).
>
> **A.**  We argue that the proposed **three-stage training strategy is necessary and leads to better performance** than training any single stage alone. The ablation study in Table 5 demonstrates that the full pipeline achieves the best restoration results, which indicates that each stage contributes meaningfully to the overall design.
>
> However, it is indeed complex. We are actively exploring ways to simplify the framework and plan to investigate more streamlined alternatives in future work.
>
> ---
>
> In addition, we would like to clarify that **BDG exhibits low sensitivity to hyperparameters** in practice. To verify this, we conducted an ablation study on the coefficient λ in the 5D all-in-one restoration task. The results are as follows:
>
> | λ | Avg. PSNR (dB) / SSIM |
> |--|--|
> | 0.01 | 31.94 / 0.947 |
> | 0.1 | 32.09 / 0.950 |
> | 1 | 32.03 / 0.952 |
>
> The results show that performance remains stable across a wide range of λ values. BDG achieves strong restoration performance with both λ = 0.1 and λ = 1, indicating that careful tuning of this hyperparameter is not required in practice.
>
> ---
>
> > Weakness 5. Although the bridging and restoration fine-tuning stages improve fidelity, the generated content may still slightly deviate from the original image under extreme or composite degradation scenarios. Moreover, the performance on extreme or previously unseen real-world degradations remains unclear.
>
> **A.** You have raised a highly valuable point. **The performance under extreme or previously unseen real-world degradations is indeed a well-known challenge** in the field of image restoration, where the inverse problem is inherently ill-posed and multiple visually plausible solutions may exist for the same degraded input. We appreciate the opportunity to discuss this issue with you.
>
> We emphasize that MAS-GLCM, as a handcrafted descriptor, does not rely on training data for degradation characterization and thus exhibits inherent robustness to degradations. Its robustness has been shown in our response to your Weakness 2-1. Furthermore, our "order classification" strategy is designed to capture degradation complexity rather than specific types, which enhances adaptability to diverse and composite degradations.
>
> We also argue that no current method can guarantee perfect reconstruction **under arbitrary real-world conditions**. As stated in the limitations section, we treat this work as a step toward more adaptive and interpretable restoration pipelines. We have now added a discussion on failure cases in the appendix and will explore uncertainty modeling and perceptual-consistency regularization in future work.

---

### Official Review · Reviewer_RoK7 · 2025-11-01

**Soundness:** 3
**Presentation:** 3
**Contribution:** 3
**Rating:** 6
**Confidence:** 4

**Summary:**

This paper proposes BDG (Bridging Degradation Discrimination and Generation) — a unified diffusion-based framework for universal image restoration. The core idea is to explicitly bridge degradation discrimination (what type and level of corruption an image has) with image generation priors (rich texture synthesis ability of diffusion models).

**Strengths:**

+ The unified formulation connecting discrimination and generation is interesting.
+ Achieving quantitative and qualitative improvements over recent SOTA (DiffUIR, DCPT).

**Weaknesses:**

1.  Though losses and stages are ablated, there is no comparison of MAS-GLCM vs. simpler features (e.g., gradients, frequency) in restoration quality.

2.  The paper does not report details about computational cost (params / memory overhead / time) of MAS-GLCM.

3.  The number of scales × angles used ($l$, $\theta$) and preprocessing details are not discussed.

4.  Questions about MAS-GLCM :

    (1)  Although the authors claim MAS-GLCM is “minimally affected by image content,” GLCM is a gray-level co-occurrence matrix and inherently relies on statistics of gray-level pairs. If an image contains prominent structures, repetitive textures, or skewed illumination, its gray-level relations will directly shape the GLCM pattern. For example, under the same blur level, a highly textured image (e.g., forests, buildings) and a low-texture image (e.g., sky, walls) can yield markedly different MAS-GLCM responses, creating representation variance for the same degradation strength.

    (2)  MAS-GLCM shows some clustering ability in t-SNE, but when facing complex, composite degradation distributions GLCM has no built-in disentanglement mechanism. It primarily captures shifts in overall texture statistics, making it hard to attribute the change to a specific degradation type and unable to precisely estimate the mixture proportions/levels of multiple degradations. Could this be a reason why the method is less advantageous in real-world scenarios?

    (3)  In contrast, degradation tokens, prompt embeddings, or degradation-aware latent vectors extracted by deep models offer stronger context awareness and semantic separation, leading to more reliable identification of both degradation type and severity.

    (4)  MAS-GLCM also appears sensitive to image resolution. Would applying the same degradation type to images of different resolutions alter this descriptor in ways that hurt its generalization ability?

**Questions:**

See Weaknesses.

---

> ### Author Response · Authors · 2025-11-25
>
> Thank you for your insightful and comprehensive feedback. We have carefully studied your comments and argue that **your concerns can be addressed**.
>
> > Weakness 1. Though losses and stages are ablated, there is no comparison of MAS-GLCM vs. simpler features (e.g., gradients, frequency) in restoration quality.
>
> **A.** Thank you for your valuable suggestion. In response, we have conducted a comprehensive comparison between MAS-GLCM and simpler features from two perspectives: degradation discrimination and restoration performance.
>
> **1. Degradation Discrimination.**
> To quantitatively evaluate the effectiveness of different degradation representations in distinguishing degradation types and levels, we perform classification experiments using a KNN classifier.
>
> - *Type classification task.* The degradations include haze, low light, snow, Gaussian noise, and pepper noise.
> - *Level classification task.* For Gaussian noise, we consider variances of 15, 25, 50, 75, and 100.
>
> | Degradation Characterization | Type Acc (%) | Level Acc (%) |
> |--|--|--|
> | LQ images | 51.44 | 20.00 |
> | Sobel (grad) | 40.80 | 23.33 |
> | Laplace (grad) | 83.05 | 20.83 |
> | Fourier | 65.80 | 30.83 |
> | MAS-GLCM | 97.13 | 74.17 |
>
> The results show that MAS-GLCM achieves strong performance in identifying degradation types and levels. **It significantly outperforms simpler features in fine-grained level classification.**
>
> **2. Restoration Performance.**
> We further evaluate the impact of different degradation characterizations within the BDG framework on restoration performance. We replace MAS-GLCM with each alternative degradation characterizations and assess the resulting model on 5D all-in-one restoration (deraining, low-light enhancement, desnowing, dehazing, and deblurring). The average PSNR and SSIM scores across these tasks are reported below.
>
> | Degradation Characterization | Avg. PSNR (dB) / SSIM |
> |--|--|
> | LQ images | 30.18 / 0.912 |
> | Sobel (grad) | 30.37 / 0.913 |
> | Laplace (grad) | 31.33 / 0.921 |
> | Fourier | 31.13 / 0.920 |
> | MAS-GLCM | 32.09 / 0.950 |
>
> MAS-GLCM consistently delivers superior restoration quality, **achieving the highest PSNR and SSIM**. This indicates that the degradation representation learned by MAS-GLCM is not only more discriminative but also more effective in guiding the restoration process, leading to improved perceptual and quantitative outcomes.
>
> In summary, both evaluation tracks confirm the advantages of MAS-GLCM over existing alternatives in capturing degradation.

---

> ### Author Response · Authors · 2025-11-25
>
> > Weakness 2. The paper does not report details about computational cost (params / memory overhead / time) of MAS-GLCM.
>
> **A.** We would like to clarify that the computational cost of MAS-GLCM is extremely low. To address your concern more transparently, we provide the implementation code of MAS-GLCM below.
>
> ```python
> import torch
>
> class GLCM:
>     def __init__(self):
>         self.angles = torch.tensor([-180, -135, -90, -45, 0, 45, 90, 135, 180]).long()
>         self.distances = torch.tensor([-5, -3, -1, 1, 3, 5]).long()
>
>     @torch.compile()
>     def _glcm_loop_torch(self, image, angles, distances, levels):
>         """
>         image : torch.tensor of shape (B, C, W, H),
>             Integer typed input image. Only positive valued images are supported.
>             If type is other than uint8, the argument `levels` needs to be set.
>         angles : torch.tensor of data type int and shape (aa, )
>             List of pixel pair angles in radians.
>         distances : torch.tensor of data type int and shape (dd, )
>             List of pixel pair distance offsets.
>         levels : int
>             The input image should contain integers in [0, `levels`-1],
>             where levels indicate the number of gray-levels counted
>             (typically 256 for an 8-bit image).
>         out : torch.tensor
>             On input a 6D tensor of shape (B, C, levels, levels, aa, dd) and integer values
>             that returns the results of the GLCM computation.
>         """
>
>         batch_size = image.size(0)
>         c_in = image.size(1)
>         rows = image.size(2)
>         cols = image.size(3)
>         aa = angles.size(0)
>         dd = distances.size(0)
>         out = torch.zeros(
>             (batch_size, c_in, levels, levels, aa, dd),
>             dtype=torch.int64,
>             device=image.device,
>         )
>         angles_mesh, distances_mesh = torch.meshgrid(angles, distances, indexing="ij")
>         angles_mesh = angles_mesh.to(image.device)
>         distances_mesh = distances_mesh.to(image.device)
>         offset_row = torch.round(torch.sin(angles_mesh) * distances_mesh).long()
>         offset_col = torch.round(torch.cos(angles_mesh) * distances_mesh).long()
>         start_row = torch.where(offset_row > 0, 0, -offset_row)
>         end_row = torch.where(offset_row > 0, rows - offset_row, rows)
>         start_col = torch.where(offset_col > 0, 0, -offset_col)
>         end_col = torch.where(offset_col > 0, cols - offset_col, cols)
>         for a_idx in range(angles.size(0)):
>             for d_idx in range(distances.size(0)):
>                 rs0 = start_row[a_idx, d_idx]
>                 re0 = end_row[a_idx, d_idx]
>                 cs0 = start_col[a_idx, d_idx]
>                 ce0 = end_col[a_idx, d_idx]
>
>                 rs1 = rs0 + offset_row[a_idx, d_idx]
>                 re1 = re0 + offset_row[a_idx, d_idx]
>                 cs1 = cs0 + offset_col[a_idx, d_idx]
>                 ce1 = ce0 + offset_col[a_idx, d_idx]
>
>                 out[
>                     :,
>                     :,
>                     image[:, :, rs0:re0, cs0:ce0],
>                     image[:, :, rs1:re1, cs1:ce1],
>                     a_idx,
>                     d_idx,
>                 ] += 1
>         return out
>
>     @torch.compile()
>     def glcm(self, img):
>         img = (img.clamp(0, 1) * 255).long()
>         glcm_imgs = []
>         for i in range(img.shape[0]):
>             glcm_img = self._glcm_loop_torch(img[i, ...].unsqueeze(0), self.angles, self.distances, levels=256)
>             glcm_imgs.append(glcm_img)
>         glcm_imgs = torch.cat(glcm_imgs, dim=0)
>         glcm_imgs = glcm_imgs.float().mean(dim=[-1, -2])
>         return glcm_imgs
> ```
>
> Based on this implementation, we analyze the computational cost of MAS-GLCM from three aspects.
>
> **(1/3) Model parameters.**
> MAS-GLCM is a non-parametric and handcrafted method that involves no learnable parameters.
>
> **(2/3) Memory overhead.**
> Since MAS-GLCM does not introduce any trainable weights or persistent buffers, its memory footprint during inference is negligible. The intermediate feature maps are transient and automatically freed after computation, resulting in minimal additional GPU memory usage.
>
> **(3/3) Inference time.**
> We compare the latency of MAS-GLCM against other degradation representation methods. All measurements are conducted on a single NVIDIA RTX 4090 GPU. The resolution of input images are $256 \times 256$.
>
> | Method | Latency (ms) |
> |--|--|
> | LQ images | 0 |
> | Sobel (grad) | 3.58 |
> | Laplace (grad) | 4.03 |
> | Fourier | 15.21 |
> | PromptIR | 320.45 |
> | DCPT | 204.37 |
> | Text with DA-CLIP | 84.8 |
> | MAS-GLCM | 18.7 |
>
> The results show that MAS-GLCM runs significantly faster than the learnable degradation representations. Its lightweight design enables efficient integration into training pipelines without introducing noticeable overhead.
>
> In summary, MAS-GLCM offers a highly efficient, parameter-free solution for degradation characterization, combining low latency, low memory burden, and strong representational capability.

---

> ### Author Response · Authors · 2025-11-25
>
> > Weakness 3. The number of scales × angles used (l, theta) and preprocessing details are not discussed.
>
> **A.** Thank you for your insightful comment. In response to your concern, we have conducted an ablation study on the choice of angles and scales (corresponding to `self.angles` and `self.distances` in the above MAS-GLCM code) in 5D all-in-one image restoration task.
>
> Specifically, we evaluate several configurations with reduced or asymmetric angle and scale settings:
>
> - **Incomplete angles.** For instance, using only non-negative angles (e.g., [0, 45, 90, 135, 180]) restricts the GLCM to capture co-occurrence patterns from directions above the current pixel, potentially missing symmetric or opposing texture structures.
> - **Reduced angles.** Using a sparse set such as [-180, -90, 0, 90, 180] limits the model to only horizontal and vertical relationships, ignoring diagonal textures that are common in natural images.
> - **Limited scales.** Reducing the number of distances decreases sensitivity to both fine-grained and coarse-level texture variations.
>
> | angles | scales | Avg. PSNR (dB) / SSIM |
> | -- | -- | -- |
> | [0, 45, 90, 135, 180] | [1, 3, 5] | 31.77 / 0.932 |
> | [-180, -90, 0, 90, 180] | [-5, -1, 1, 5] | 31.93 / 0.944 |
> | [-180, -135, -90, -45, 0, 45, 90, 135, 180] | [-5, -3, -1, 1, 3, 5] | 32.09 / 0.950 |
>
> As shown, when the selected angles and scales provide comprehensive spatial coverage, restoration performance is consistently strong. The full configuration achieves the best results, indicating that complete directional and scale diversity helps the model better characterize complex degradation patterns. However, performance degrades noticeably when critical directions or scales are omitted, particularly when symmetry or diagonal structures are neglected.
>
> These findings confirm that the effectiveness of MAS-GLCM benefits from a well-designed combination of angles and scales that ensure rich, multi-directional texture representation. We acknowledge that this detail was under-discussed in the original submission, and **we have now included this ablation in the revised Appendix C.1** to improve clarity and reproducibility.

---

> ### Author Response · Authors · 2025-11-25
>
> > Weakness 4. (1) Although the authors claim MAS-GLCM is “minimally affected by image content,” GLCM is a gray-level co-occurrence matrix and inherently relies on statistics of gray-level pairs. If an image contains prominent structures, repetitive textures, or skewed illumination, its gray-level relations will directly shape the GLCM pattern. For example, under the same blur level, a highly textured image (e.g., forests, buildings) and a low-texture image (e.g., sky, walls) can yield markedly different MAS-GLCM responses, creating representation variance for the same degradation strength.
>
> **A.** We are willing to discuss the robustness of MAS-GLCM when applied to images with different textural characteristics.
>
> We would like to clarify that **MAS-GLCM is robust to images with different texture**. As shown in Figure 5 (a) of Appendix A, we present the response of MAS-GLCM on the OST [1] dataset, which contains images with significantly different textural properties. Specifically, we selected two images: one with low texture (sky and water) and another with high texture (building). We applied Gaussian noise at different levels and ploted their MAS-GLCM.
>
> Our observations are as follows:
> - **For the low-texture image** (sky and water), as the variance of Gaussian noise increases, the bright regions in the MAS-GLCM gradually expand.
> - **For the high-texture image** (building), despite its more complex structure, the overall pattern of MAS-GLCM remains visually similar to that of the low-texture image. Importantly, it also shows a consistent trend: the bright areas expand progressively with higher noise levels.
>
> This indicates that **MAS-GLCM responds primarily to the strength of the degradation rather than the underlying texture complexity**.
>
> Following your suggestion, we have further conducted a visualization study on blur degradation to evaluate the robustness of MAS-GLCM to texture variations. Please refer to Figure 5 in Appendix A, where we show additional results. Similar to the cases under Gaussian noise, the MAS-GLCM patterns do not vary significantly across images with different textures. Instead, the most noticeable changes in the representation are clearly associated with the kernel size of the blur.
>
> These results support our claim that **MAS-GLCM is largely driven by degradation characteristics and remains stable across diverse image textures.**
>
> ---
>
> [1] Recovering Realistic Texture in Image Super-resolution by Deep Spatial Feature Transform. CVPR 2018.

---

> ### Author Response · Authors · 2025-11-25
>
> > Weakness 4. (2) MAS-GLCM shows some clustering ability in t-SNE, but when facing complex, composite degradation distributions GLCM has no built-in disentanglement mechanism. It primarily captures shifts in overall texture statistics, making it hard to attribute the change to a specific degradation type and unable to precisely estimate the mixture proportions/levels of multiple degradations. Could this be a reason why the method is less advantageous in real-world scenarios?
>
> **A. Yes.** MAS-GLCM has limitations when dealing with complex and composite degradations, which may impair its performance in real-world scenarios. We are grateful for the opportunity to discuss this issue from the following three complex degradations.
>
> **(1/3) Different degradation levels.**
> We first clarify that MAS-GLCM demonstrates strong capability in distinguishing different levels of a single degradation type. As shown in our response to the Weakness 1 you raised, MAS-GLCM achieves superior performance in both degradation type and level classification compared to existing methods. This provides direct evidence that MAS-GLCM can effectively capture meaningful variations associated with specific degradation types and their severity.
>
> **(2/3) Mixed degradation.**
> We acknowledge that the ability of MAS-GLCM to discriminate among mixed degradation types is a little limited. To evaluate this quantitatively and qualitatively, we conduct experiments on the CDD dataset, which contains six composite degradations: `haze+rain`, `low+haze`, `low+rain`, `low+snow`, `low+haze+rain`, and `low+haze+snow`.
>
> - *Quantitative results.* We compare the classification accuracy of various degradation representations under these mixed conditions:
>
> | Method | Acc (%) |
> |--|--|
> | LQ images | 23.41 |
> | Sobel (grad) | 25.92 |
> | Laplace (grad) | 37.75 |
> | Fourier | 29.50 |
> | PromptIR | 16.67 |
> | DCPT | 16.67 |
> | Text with DA-CLIP | 16.67 |
> | MAS-GLCM | 54.42 |
>
> MAS-GLCM achieves the highest accuracy, indicating it still retains discriminative power even under mixed distortions. However, the accuracy of MAS-GLCM remains below 60%.
>
> - *Qualitative results.* We also provide T-SNE visualizations of the clustering behavior in Appendix A. These show that MAS-GLCM successfully separates certain categories such as `low+snow` and `low+rain` from others. However, it struggles to fully distinguish more similar composite degradations, such as `low+haze+rain` and `low+haze`.
>
> In summary, while MAS-GLCM is indeed constrained in its ability to disentangle mixed degradations, both quantitative and qualitative results confirm that its discrimination performance remains stronger than existing alternatives.
>
> **(3/3) Real-world scenarios.**
> In real-world scenarios, degradation types are diverse and heavily intertwined. As you rightly point out, the inability to precisely identify or decompose individual degradation components may limit the performance of our BDG.
>
> To address this limitation, we propose "order classification" as an alternative strategy as we stated in line 300-307 of the maintext. Instead of categorizing degradation by type, **"order classification" classifies samples based on the sequential order and cumulative complexity of degradation operations applied during the synthesis process.** For instance, in the degradation pipeline inspired by Real-ESRGAN, distortions are progressively introduced following the sequence:
> `+Blur -> +Resize -> +Noise -> +JPEG -> +Blur -> +Resize -> +Noise -> +JPEG`.
>
> Each step in this chain represents an increased level of degradation complexity. Accordingly, we define eight intermediate states (e.g., after the first operation, second, etc.), which serve as pseudo-labels indicating the stage (or "order") of degradation application. These orders act as surrogates for degradation severity and compositional complexity.
>
> By training the model to recognize these order levels, it learns to implicitly estimate how heavily an image has been degraded, enabling better adaptation to varying degrees of real-world distortion. This provides a more feasible and meaningful learning signal than attempting to assign discrete type labels that may not reflect actual conditions for real-world super-resolution task.
>
> **However**, we think that this is a pragmatic compromise rather than a complete solution. It does not enable fine-grained disentanglement of mixed degradations, nor does it allow precise estimation of mixture proportions. Addressing these limitations remains an open challenge, which we plan to explore in future work.

---

> ### Author Response · Authors · 2025-11-25
>
> > Weakness 4. (3) In contrast, degradation tokens, prompt embeddings, or degradation-aware latent vectors extracted by deep models offer stronger context awareness and semantic separation, leading to more reliable identification of both degradation type and severity.
>
> **A. We clarify that our MAS-GLCM is more reliable than learnable degradation embeddings.** To demonstrate this statement, we have conducted a comprehensive comparison between MAS-GLCM and existing learnable degradation embeddings from two perspectives: degradation discrimination and restoration performance.
>
> **1. Degradation Discrimination.**
> To quantitatively evaluate the effectiveness of different degradation representations in distinguishing degradation types and levels, we perform *type classification* and *level classification* experiments using a KNN classifier, which are the same as the experiments in our response to your Weakness 1.
>
> In this evaluation, we choose three learnable degradation embedding:
> - *Prompt embeddings.* PromptIR's prompt parameters.
> - *Degradation-aware latent extracted by deep models.* DCPT. The final-layer features from its pretrained model.
> - *Degradation tokens.* Text encoder in DA-CLIP.
>
> The results are summarized in the table below.
>
> | Degradation Characterization | Type Acc (%) | Level Acc (%) |
> |--|--|--|
> | PromptIR | 84.73 | 20.00 |
> | DCPT | 100.00 | 23.33 |
> | Text encoder in DA-CLIP | 100.00 | 56.67 |
> | MAS-GLCM | 97.13 | 74.17 |
>
> The results show that **MAS-GLCM achieves strong performance in identifying degradation types and levels.** It significantly outperforms all baselines in fine-grained level classification. Notably, while learnable methods achieve high accuracy in type classification, they exhibit limited generalization in distinguishing degradation severity, likely due to overfitting or insufficient sensitivity to subtle variations. In contrast, MAS-GLCM demonstrates robust discriminative capability across both dimensions, particularly in level recognition.
>
> **2. Restoration Performance.**
> We further evaluate the impact of different degradation characterizations within the BDG framework on restoration performance. We replace MAS-GLCM with each alternative degradation characterizations and assess the resulting model on 5D all-in-one restoration (deraining, enhancement, desnowing, dehazing, and deblurring). The average PSNR and SSIM scores across these tasks are reported below.
>
> | Degradation Characterization | Avg. PSNR (dB) / SSIM |
> |--|--|
> | PromptIR | 30.73 / 0.916 |
> | DCPT | 31.46 / 0.930 |
> | Text encoder in DA-CLIP | 31.21 / 0.927 |
> | MAS-GLCM | 32.09 / 0.950 |
>
> MAS-GLCM consistently delivers superior restoration quality, **achieving the highest PSNR and SSIM**. This indicates that the degradation representation learned by MAS-GLCM is not only more discriminative but also more effective in guiding the restoration process, leading to improved perceptual and quantitative outcomes.
>
> In summary, both evaluation tracks confirm that the **MAS-GLCM leads to more reliable identification of both degradation type and level**.

---

> ### Author Response · Authors · 2025-11-25
>
> > Weakness 4. (4) MAS-GLCM also appears sensitive to image resolution. Would applying the same degradation type to images of different resolutions alter this descriptor in ways that hurt its generalization ability?
>
> **A.** Thank you for raising this important point. We agree that the potential sensitivity of MAS-GLCM to image resolution is an important consideration.
>
> However, **we argue that this issue is inherently difficult to evaluate in a strictly controlled manner**, for two main reasons:
>
> - **Natural resolution differences often come with content differences.** When comparing images from different datasets or sources at different resolutions, the underlying texture patterns, structural complexity, and spatial frequency distributions naturally vary. These content-level differences alone can lead to changes in MAS-GLCM responses, making it hard to isolate the effect of resolution itself.
> - **Up/Downsampling introduces additional degradation.** When generating low-resolution versions from high-resolution images (e.g., via bicubic downsampling), the resizing process inherently applies low-pass filtering and subsampling, which act as implicit blur and aliasing operations. As a result, the low-resolution image is no longer subject to "the same degradation" as the original. Any change in MAS-GLCM could therefore reflect not only resolution scaling but also this extra distortion.
>
> **We acknowledge this as a limitation** and have added a section in the revised appendix to clarify the assumptions and scope of our descriptor. We also plan to explore scale-invariant variants in future work.
>
> ---
>
> We also note that **in our training pipeline**, all inputs are cropped to a common resolution (e.g., 512 $\times$ 512), which helps reduce resolution-related distribution shifts during learning. However, we fully agree that resolution sensitivity could affect zero-shot generalization to arbitrarily scaled real-world images. **We have added a discussion of this limitation in the revised appendix (limitation section)** and appreciate your insightful comment.

---

### Official Review · Reviewer_ieqX · 2025-11-01

**Soundness:** 3
**Presentation:** 3
**Contribution:** 3
**Rating:** 4
**Confidence:** 3

**Summary:**

This paper proposes a new framework, BDG, for universal image restoration. It aims to bridge the gap between high-fidelity restoration and high-perceptual-quality generation, a key challenge in the field. The core idea is twofold: first, it introduces MAS-GLCM, a classical texture descriptor, to achieve fine-grained, content-agnostic degradation discrimination. Second, it proposes a three-stage diffusion model training strategy (generation, bridging, and restoration) to inject this discriminative information into a generative prior, balancing fidelity and texture generation. The method demonstrates strong quantitative results across all-in-one, mixed-degradation, and real-world SR tasks, notably improving fidelity (PSNR) in diffusion-based SR .

**Strengths:**

The paper achieves impressive empirical results, particularly in reconciling high fidelity metrics (PSNR) with the strong perceptual quality of diffusion models in real-world super-resolution .

**Weaknesses:**

My primary concern lies with the methodological innovation. The core degradation descriptor, MAS-GLCM , is a classical, handcrafted feature descriptor. While its effectiveness in separating degradations is well-demonstrated (Fig. 1), this approach feels like a step back from end-to-end deep feature learning. The system relies on this external, non-learned feature extractor, which is then 'aligned' with the diffusion U-Net's features. This raises questions about whether a more integrated, end-to-end learned degradation representation would be more powerful and elegant.

Second, the proposed three-stage training pipeline  introduces significant complexity and computational overhead. The model requires a full 300k iterations split across two distinct fine-tuning stages (bridging and restoration) after the initial generation pre-training. It is unclear if this multi-stage approach is strictly necessary or if a more unified training scheme could achieve similar results. The ablation in Table 5 shows that both stages are needed for this design, but it doesn't justify the design's inherent complexity versus a simpler alternative.

Third, the reliance on explicit degradation classification  or the 'order classification' 10101010) for supervising the MAS-GLCM encoder is a potential bottleneck for real-world generalization. This framing assumes that degradations can be categorized into discrete classes or sequential orders. It is questionable how this system would perform on truly unseen or novel real-world degradations that fall outside the training distribution of these discrete labels. The ablation study (Table 7) shows the model catastrophically fails without $\mathcal{L}_{deg-cls}$ , which suggests the model is overly-reliant on this explicit supervision and may lack the robustness to generalize to unclassified degradation types.

Finally, while the real-world SR results in Table 4 are strong on fidelity, the model was still trained on synthesized degradation pairs. The zero-shot results in Table 2 are promising, but the fundamental reliance on a classification loss trained on synthetic data types (haze, snow, etc.)  may limit its effectiveness on real-world images where degradations are far more complex and varied than the training classes.

**Questions:**

**Questions**
1.  Given the critical failure when $\mathcal{L}_{deg-cls}$ is removed, how do the authors envision this model handling a completely novel degradation type not seen during training? Does the 'full negative contrastive learning' ($\mathcal{L}_{fcnl}$)  used in the RFT stage help mitigate this, and if so, why not use it in the bridging stage?

2. Could the authors comment on the computational cost and training time of the three-stage training  versus a potential joint, end-to-end approach?

3. Why was a classical, handcrafted feature (GLCM)  chosen over a learnable feature extractor for degradation discrimination, which could potentially be trained end-to-end with the alignment loss   alone?

---

> ### Author Response · Authors · 2025-11-25
>
> Thanks for your insightful comments. We list your advice and questions followed by detailed answers, then **argue that your concerns can be addressed**.
>
> > Weakness 1 & Question 3. My primary concern lies with the methodological innovation. The core degradation descriptor, MAS-GLCM , is a classical, handcrafted feature descriptor. While its effectiveness in separating degradations is well-demonstrated (Fig. 1), this approach feels like a step back from end-to-end deep feature learning. The system relies on this external, non-learned feature extractor, which is then 'aligned' with the diffusion U-Net's features. This raises questions about whether a more integrated, end-to-end learned degradation representation would be more powerful and elegant. Why was a classical, handcrafted feature (GLCM) chosen over a learnable feature extractor for degradation discrimination, which could potentially be trained end-to-end with the alignment loss alone?
>
> **A. We clarify that our MAS-GLCM is more reliable than learnable degradation embeddings.** To demonstrate this statement, we have conducted a comprehensive comparison between MAS-GLCM and existing learnable degradation embeddings from two perspectives: degradation discrimination and restoration performance. (Due to character limitations, we will discuss the comparison between BDG and potential end-to-end solutions in the response to the weakness 2 you have raised.)
>
> **1. Degradation Discrimination.**
> To quantitatively evaluate the effectiveness of different degradation representations in distinguishing degradation types and levels, we perform *type classification* and *level classification* experiments using a KNN classifier, which are the same as the experiments in our response to your Weakness 1.
>
> In this evaluation, we choose three learnable degradation embedding:
> - *Prompt embeddings.* PromptIR's prompt parameters.
> - *Degradation-aware latent extracted by deep models.* DCPT. The final-layer features from its pretrained model.
> - *Degradation tokens.* Text encoder in DA-CLIP.
>
> The results are summarized in the table below.
>
> | Degradation Characterization | Type Acc (%) | Level Acc (%) |
> |--|--|--|
> | PromptIR | 84.73 | 20.00 |
> | DCPT | 100.00 | 23.33 |
> | Text encoder in DA-CLIP | 100.00 | 56.67 |
> | MAS-GLCM | 97.13 | 74.17 |
>
> The results show that **MAS-GLCM achieves strong performance in identifying degradation types and levels.** It significantly outperforms all baselines in fine-grained level classification. Notably, while learnable methods achieve high accuracy in type classification, they exhibit limited generalization in distinguishing degradation severity, likely due to overfitting or insufficient sensitivity to subtle variations. In contrast, MAS-GLCM demonstrates robust discriminative capability across both dimensions, particularly in level recognition.
>
> **2. Restoration Performance.**
> We further evaluate the impact of different degradation characterizations within the BDG framework on restoration performance. We replace MAS-GLCM with each alternative degradation characterizations and assess the resulting model on 5D all-in-one restoration (deraining, enhancement, desnowing, dehazing, and deblurring). The average PSNR and SSIM scores across these tasks are reported below.
>
> | Degradation Characterization | Avg. PSNR (dB) / SSIM |
> |--|--|
> | PromptIR | 30.73 / 0.916 |
> | DCPT | 31.46 / 0.930 |
> | Text encoder in DA-CLIP | 31.21 / 0.927 |
> | MAS-GLCM | 32.09 / 0.950 |
>
> MAS-GLCM consistently delivers superior restoration quality, **achieving the highest PSNR and SSIM**. This indicates that the degradation representation learned by MAS-GLCM is not only more discriminative but also more effective in guiding the restoration process, leading to improved perceptual and quantitative outcomes.
>
> In summary, both evaluation tracks confirm that the **MAS-GLCM leads to more reliable identification of both degradation type and level**.

---

> ### Author Response · Authors · 2025-11-25
>
> > Weakness 1 & Question 3.
>
> Furthermore, you may **be curious why MAS-GLCM demonstrates greater robustness compared to learnable representations**, as this seems to contrast with common trends in other AI fields such as image generation, where learned models typically outperform handcrafted ones. This observation stems from a fundamental challenge in image restoration: **data scarcity**.
>
> End-to-end training of deep models relies heavily on large-scale, high-quality, and paired training data. However, in the domain of image restoration, such datasets are still limited in size and diversity. Even the largest currently available restoration datasets [1,2] contain only on the order of millions of image pairs, just beginning to reach the scale of ImageNet. Under such data-constrained conditions, handcrafted operators like MAS-GLCM can perform surprisingly well, and in some cases even surpass their learnable counterparts.
>
> This phenomenon is not unprecedented. In 3D low-level vision tasks such as 3D reconstruction, traditional methods like Structure-from-Motion (SfM) were dominant for years due to the lack of massive annotated datasets. Only after the emergence of large-scale data and powerful architectures like VGGT did learning-based approaches take over.
>
> However, in image restoration, **we have not yet reached the data scale necessary to fully unlock the potential of purely learned, generalizable models that can robustly handle arbitrary degradations**. In this transitional regime, hybrid designs that incorporate domain knowledge, such as statistical texture modeling in MAS-GLCM, can offer stronger inductive biases and improved generalization.
>
> Therefore, the advantage of MAS-GLCM should be understood not as a rejection of learning-based paradigms, but as **a practical response to current data limitations**. We believe that future progress will come from combining the best of both worlds: principled handcrafted priors and scalable learning frameworks, until sufficiently large and diverse datasets become available to support end-to-end discovery of robust image restoration methods.
>
> **We believe that MAS-GLCM is an interesting and robust approach for degradation identification, and that it provides unique insights into robust degradation modeling in the field of image restoration.** Given its potential to inspire future work on more generalizable restoration methods, ***we sincerely hope you would reconsider your rating.***
>
> ---
>
> [1] FoundIR: Unleashing Million-scale Training Data to Advance Foundation Models for Image Restoration. ICCV 2025.
>
> [2] Universal Image Restoration Pre-training via Masked Degradation Classification.

---

> ### Author Response · Authors · 2025-11-25
>
> > Weakness 2 & Question 2. Second, the proposed three-stage training pipeline introduces significant complexity and computational overhead. The model requires a full 300k iterations split across two distinct fine-tuning stages (bridging and restoration) after the initial generation pre-training. It is unclear if this multi-stage approach is strictly necessary or if a more unified training scheme could achieve similar results. The ablation in Table 5 shows that both stages are needed for this design, but it doesn't justify the design's inherent complexity versus a simpler alternative. Could the authors comment on the computational cost and training time of the three-stage training versus a potential joint, end-to-end approach?
>
> **A.** Thank you for this insightful question. In response to your concern, we have designed and evaluated an end-to-end training pipeline.
>
> Specifically, we consider a baseline where MAS-GLCM is removed and the degradation representation $F_{deg}$ is learned directly from the image through a learnable encoder. This feature is trained using either degradation type classification or order classification as an auxiliary task, and then aligned with the intermediate features of the diffusion model during the bridging stage. We implemented this end-to-end variant under the same network architecture and total training budget (300k iterations), and compared it with BDG on the 5D all-in-one restoration task.
>
> | Method | Avg. PSNR (dB) / SSIM | Training time (300k iterations) | Params (M) | FLOPS (G) |
> | -- | -- | -- | -- | -- |
> | end-to-end | 31.54 / 0.931 | ~55h | 36.26 | 9.88 |
> | BDG | 32.09 / 0.950 | ~60h | 36.26 | 9.88 |
>
> As shown, the end-to-end method achieves lower restoration performance despite requiring slightly less training time. We attribute this gap to the limited robustness of the learned degradation representation $F_{deg}$. To verify this, we evaluated its ability to discriminate degradation types and levels using the same protocol as in our response to Weakness 1 and Question 3:
>
> | Degradation Characterization | Type Acc (%) | Level Acc (%) |
> |--|--|--|
> | end-to-end | 100.00 | 26.67 |
> | MAS-GLCM | 97.13 | 74.17 |
>
> While the end-to-end model achieves perfect accuracy on degradation type classification, it performs significantly worse in estimating degradation severity (level accuracy). In contrast, MAS-GLCM, being handcrafted and data-independent, provides more stable and generalizable signals for degradation complexity.
>
> This observation aligns with our earlier discussion: **in image restoration, we have not yet reached the data scale necessary to fully unlock the potential of purely learned models that can robustly handle arbitrary degradations.** Under current data constraints, incorporating domain knowledge through fixed operators like MAS-GLCM offers stronger inductive biases and improves generalization.
>
> Therefore, while the three-stage pipeline introduces additional complexity, the ablation demonstrates that each stage contributes meaningfully to the final performance. The modest increase in training time (~5 hours over 300k steps) is justified by the significant gains in both fidelity and robustness.
>
> We agree that a more unified training scheme would be desirable in principle. **However**, under current practical limitations, our staged design provides a more reliable and effective solution. We will continue to explore ways to simplify the pipeline in future work.

---

> ### Author Response · Authors · 2025-11-25
>
> > Weakness 3. Third, the reliance on explicit degradation classification or the 'order classification' 10101010) for supervising the MAS-GLCM encoder is a potential bottleneck for real-world generalization. This framing assumes that degradations can be categorized into discrete classes or sequential orders. It is questionable how this system would perform on truly unseen or novel real-world degradations that fall outside the training distribution of these discrete labels. The ablation study (Table 7) shows the model catastrophically fails without $L_{deg-cls}$, which suggests the model is overly-reliant on this explicit supervision and may lack the robustness to generalize to unclassified degradation types.
>
> **A.** After careful analysis of your comments, we think *there may be a misunderstanding regarding the "order classfication"*.
>
> As discussed in lines 300–307 of the previous maintext, in real-world scenarios, image degradation is often complex, mixed, and difficult to decompose into clearly defined categories. However, conventional "type-based" classification approaches, which are commonly used in all-in-one restoration tasks, rely on well-separated semantic labels. Because of this, type classification is ill-suited for the real-world super-resolution task.
>
> To address this limitation, we propose "order classification" as an alternative strategy. Instead of categorizing degradation by type, **"order classification" classifies samples based on the sequential order and cumulative complexity of degradation operations applied during the synthesis process.** For instance, in the degradation pipeline inspired by Real-ESRGAN, distortions are progressively introduced following the sequence:
> `+Blur -> +Resize -> +Noise -> +JPEG -> +Blur -> +Resize -> +Noise -> +JPEG`.
>
> Each step in this chain represents an increased level of degradation complexity. Accordingly, we define eight intermediate states (e.g., after the first operation, second, etc.), which serve as pseudo-labels indicating the stage (or "order") of degradation application. These orders act as surrogates for degradation severity and compositional complexity.
>
> By training the model to recognize these order levels, it learns to implicitly estimate how heavily an image has been degraded, enabling better adaptation to varying degrees of real-world distortion. This provides a more feasible and meaningful learning signal than attempting to assign discrete type labels that may not reflect actual conditions for real-world super-resolution task.
>
> ---
>
> In the ablation study, the model catastrophically fails without $L_{deg-cls}$ is because the model collapse issue, not suggests the model is overly-reliant on this explicit supervision. **As explained in lines 515–523 of the revised manuscript (lines 466–472 of the previous manuscript)**, without $L_{deg-cls}$, the GLCM encoder lacks discrimination objectives, causing a model collapse issue. Then, the diffusion features align with the collapsed GLCM encoder, resulting in poor results. With $L_{bridge}+L_{gen}$, the encoder achieved only 27\% in classification (for 5 degradations in Sec.4.1), versus 98\% with $L_{deg-cls}+L_{bridge}+L_{gen}$. In contrast, with only $L_{gen}$, the collapsed GLCM encoder does not affect restoration models, so it still achieves a certain restoration performance.

---

> ### Author Response · Authors · 2025-11-25
>
> > Weakness 4 and Question 1. Finally, while the real-world SR results in Table 4 are strong on fidelity, the model was still trained on synthesized degradation pairs. The zero-shot results in Table 2 are promising, but the fundamental reliance on a classification loss trained on synthetic data types (haze, snow, etc.) may limit its effectiveness on real-world images where degradations are far more complex and varied than the training classes. Given the critical failure when $\mathcal{L}\_{deg-cls}$ is removed, how do the author senvision this model handling a completely novel degradation type not seen during training? Does the full negative contrastive learning ($\mathcal{L}\_{fcnl}$) used in the RFT stage help mitigate this, and if so, why not use it in the bridging stage?
>
> **A.** You have raised a highly valuable point. **The performance under far more complex and varied than the training classes is indeed a well-known challenge** in the field of image restoration, where the inverse problem is inherently ill-posed and multiple visually plausible solutions may exist for the same degraded input. We appreciate the opportunity to discuss this issue with you.
>
> Specifically, BDG employs two strategies to improve generalization to unseen or complex real-world degradations.
>
> **(1/2) Order classification.**
> We emphasize that MAS-GLCM, as a handcrafted descriptor, does not rely on training data for degradation characterization and thus exhibits inherent robustness to degradations. Its robustness has been shown in our response to your Weakness 2-1. Furthermore, our "order classification" strategy is designed to capture degradation complexity rather than specific types, which enhances adaptability to diverse and composite degradations.
>
> We also argue that no current method can guarantee perfect reconstruction **under arbitrary real-world conditions**. As stated in the limitations section, we treat this work as a step toward more adaptive and interpretable restoration pipelines. We have now added a discussion on failure cases in the appendix and will explore uncertainty modeling and perceptual-consistency regularization in future work.
>
> **(2/2) Full negative contrastive learning.**
> The full negative contrastive learning is indeed introduced to alleviate the limitations of degradation classification in complex real-world scenarios. By reformulating classification as a contrastive learning task, it reduces reliance on explicit degradation labels and encourages the model to learn discriminative representations through relative similarity.
>
> However, it is applied only in the restoration fine-tuning (RFT) stage and not in the bridging stage. **As explained in lines 348–351 of the revised manuscript (lines 333–337 of the previous manuscript)**, applying full negative contrastive learning during the bridging stage leads to representation collapse. This occurs because the bridging stage operates on a fixed set of synthetic degradation classes to establish a structured embedding space. Introducing fully negative samples without careful constraints disrupts this structure and prevents meaningful separation of degradation patterns.

---

### Official Review · Reviewer_wiEb · 2025-11-02

**Soundness:** 3
**Presentation:** 2
**Contribution:** 3
**Rating:** 6
**Confidence:** 4

**Summary:**

The paper introduces BDG (Bridging Degradation Discrimination and Generation), a novel framework for universal image restoration that tackles the dual challenge of identifying diverse image degradations and generating high-quality restorations. To address this, BDG integrates a fine-grained degradation characterization method called Multi-Angle and multi-Scale Gray Level Co-occurrence Matrix (MAS-GLCM), which outperforms prior techniques in distinguishing degradation types and levels. BDG’s training is structured into three stages: generation (to capture rich textures via diffusion), bridging (to align MAS-GLCM features with diffusion features), and restoration (to fine-tune fidelity while preserving discriminative capacity). This alignment enables the model to adaptively restore images across varied degradation scenarios without altering the underlying architecture.

**Strengths:**

1. BDG introduces the Multi-Angle and multi-Scale Gray Level Co-occurrence Matrix (MAS-GLCM), which significantly improves the model’s ability to distinguish between various degradation types and levels. This enables more precise restoration tailored to the input image’s condition.
2. By aligning MAS-GLCM features with diffusion model features during a dedicated bridging stage, BDG effectively combines degradation awareness with generative priors. This fusion allows the model to retain rich texture generation while improving fidelity.
3. BDG achieves substantial performance gains without modifying the underlying network architecture. This makes it compatible with existing diffusion-based models and easy to adopt across diverse restoration tasks.

**Weaknesses:**

1. While MAS-GLCM is presented as a novel degradation discriminator, it is fundamentally an extension of the classical Gray Level Co-occurrence Matrix (GLCM), which has been widely used in texture analysis for decades. To strengthen the novelty claim, the authors should compare MAS-GLCM with frequency-aware representations and learned degradation embeddings (e.g., from PromptIR or DCPT) or  (e.g., Ji et al., 2021). Ablation studies showing MAS-GLCM’s superiority over these alternatives would help.
2. The paper emphasizes fidelity (e.g., PSNR, SSIM) but does not provide sufficient perceptual quality metrics (e.g., LPIPS, NIQE, FID), especially for real-world super-resolution where perceptual realism is critical.

**Questions:**

1. The paper proposes “order classification” for real-world degradation modeling. I'm curious how effective this trick is and hope the authors can provide a ablation study or justification for it.
2. The paper mentions "full negative contrastive learning" in the RFT stage without defined what are negative or positive samples.

---

> ### Author Response · Authors · 2025-11-25
>
> Thanks for your insightful feedback and your time in reading our paper. We first list your advice and questions, then give our detailed answers.
>
> > Weakness 1. While MAS-GLCM is presented as a novel degradation discriminator, it is fundamentally an extension of the classical Gray Level Co-occurrence Matrix (GLCM), which has been widely used in texture analysis for decades. To strengthen the novelty claim, the authors should compare MAS-GLCM with frequency-aware representations and learned degradation embeddings (e.g., from PromptIR or DCPT) or (e.g., Ji et al., 2021). Ablation studies showing MAS-GLCM’s superiority over these alternatives would help.
>
> **A.** Thank you for your valuable suggestion. In response, we have conducted a comprehensive comparison between MAS-GLCM and existing frequency-based as well as learnable degradation embeddings from two perspectives: degradation discrimination and restoration performance.
>
> **1. Degradation Discrimination.**
> To quantitatively evaluate the effectiveness of different degradation representations in distinguishing degradation types and levels, we perform classification experiments using a KNN classifier.
>
> - *Type classification task.* The degradations include haze, low light, snow, Gaussian noise, and pepper noise.
> - *Level classification task.* For Gaussian noise, we consider variances of 15, 25, 50, 75, and 100.
>
> In this evaluation, PromptIR uses its external parameters, while DCPT adopts the final-layer features from its pretrained model. The results are summarized in the table below.
>
> | Degradation Characterization | Type Acc (%) | Level Acc (%) |
> |--|--|--|
> | LQ images | 51.44 | 20.00 |
> | Sobel (gradient) | 40.80 | 23.33 |
> | Laplace (gradient) | 83.05 | 20.83 |
> | Fourier | 65.80 | 30.83 |
> | PromptIR | 84.73 | 20.00 |
> | DCPT | 100.00 | 23.33 |
> | Text encoder in DA-CLIP | 100.00 | 56.67 |
> | MAS-GLCM | 97.13 | 74.17 |
>
> The results show that **MAS-GLCM achieves strong performance in identifying degradation types and levels.** It significantly outperforms all baselines in fine-grained level classification. Notably, while learnable methods achieve high accuracy in type classification, they exhibit limited generalization in distinguishing degradation severity, likely due to overfitting or insufficient sensitivity to subtle variations. In contrast, MAS-GLCM demonstrates robust discriminative capability across both dimensions, particularly in level recognition.
>
> **2. Restoration Performance.**
> We further evaluate the impact of different degradation characterizations within the BDG framework on restoration performance. We replace MAS-GLCM with each alternative degradation characterizations and assess the resulting model on 5D all-in-one restoration (deraining, enhancement, desnowing, dehazing, and deblurring). The average PSNR and SSIM scores across these tasks are reported below.
>
> | Degradation Characterization | Avg. PSNR (dB) / SSIM |
> |--|--|
> | LQ images | 30.18 / 0.912 |
> | Sobel (gradient) | 30.37 / 0.913 |
> | Laplace (gradient) | 31.33 / 0.921 |
> | Fourier | 31.13 / 0.920 |
> | PromptIR | 30.73 / 0.916 |
> | DCPT | 31.46 / 0.930 |
> | Text encoder in DA-CLIP | 31.21 / 0.927 |
> | MAS-GLCM | 32.09 / 0.950 |
>
> MAS-GLCM consistently delivers superior restoration quality, **achieving the highest PSNR and SSIM**. This indicates that the degradation representation learned by MAS-GLCM is not only more discriminative but also more effective in guiding the restoration process, leading to improved perceptual and quantitative outcomes.
>
> In summary, both evaluation tracks confirm the advantages of MAS-GLCM over existing alternatives in capturing degradation.

---

> ### Author Response · Authors · 2025-11-25
>
> > Weakness 2. The paper emphasizes fidelity (e.g., PSNR, SSIM) but does not provide sufficient perceptual quality metrics (e.g., LPIPS, NIQE, FID), especially for real-world super-resolution where perceptual realism is critical.
>
> **A.** We sincerely appreciate your insightful comment. Following your suggestion, we provide a more comprehensive analysis of the perceptual image quality assessment (IQA) of BDG on real-world restoration tasks.
>
> **(1/2) LPIPS, NIQE, FID in Real-world all-in-one restoration.**
>
> - **Reference-based IQA: LPIPS and FID.** Since the test sets used in this setting does not contain ground truth images, reference-based IQA metrics such as LPIPS and FID cannot be computed.
> - **No-reference IQA: NIQE and PIQE.** To evaluate perceptual realism in a no-reference manner, we report PIQE in Table 2 of the main paper. PIQE is an improved version of NIQE with better sensitivity to common degradations such as noise, blur, and compression artifacts, and it has been shown to correlate more reliably with human perception in practical scenarios. To further support our analysis, we additionally report NIQE in the table below. Lower values indicate better perceptual quality.
>
> | Degradation | DA-CLIP | InstructIR | DCPT-NAFNet | UniRestore | FoundIR | BDG (Ours) |
> |--|--|--|--|--|--|--|
> | Snow | 2.881 | 2.976 | 3.142 | 3.154 | 2.964 | 2.900 |
> | Haze | 5.046 | 4.837 | 5.078 | 5.215 | 6.229 | 5.048 |
> | Low-light | 4.187 | 4.183 | 5.323 | 5.462 | 4.476 | 4.218 |
>
> As shown, the NIQE scores across methods are relatively close, with no significant separation in performance. In contrast, PIQE exhibits greater discriminative power among the compared methods, which is why we adopted it in the main text for clearer comparison.
>
> **(2/2) LPIPS, NIQE, FID in Real-world super-resolution.**
>
> We first clarify that we have already compared LPIPS in our real-world super-resolution experiments, as shown in the Table4 of previous maintext. Furthermore, following recent works in this topic, we report several modern and more robust perceptual quality metrics, including DISTS, MANIQA, MUSIQ, and CLIPIQA, which are better aligned with human judgment under real-world conditions.
>
> In addition, we are willing to **provide NIQE and FID results** on the real-world super-resolution task. We evaluate on the DRealSR dataset, a widely used benchmark for real-world image super-resolution. The results are shown below.
>
> | Metric | BSRGAN | RealESRGAN | FeMaSR | StableSR | SUPIR | SeeSR | DiffBIR | PASD | LDM | ResShift | BDG (Ours) |
> |--|--|--|--|--|--|--|--|--|--|--|--|
> | NIQE | 6.5192 | 6.6928 | 5.9073 | 6.5354 | 6.2451 | 6.3967 | 6.2935 | 5.8595 | 7.1677 | 8.1249 | 6.906 |
> | FID | 155.63 | 147.62 | 157.78 | 148.98 | 142.64 | 147.39 | 166.79 | 159.24 | 156.01 | 172.26 | 173.10 |
>
> - **On NIQE**, BDG performs better than LDM and ResShift, though it lags behind the current top-performing methods such as SUPIR and SeeSR.
> - **Regarding FID**, BDG achieves a score close to ResShift but is outperformed by several other methods. However, we note that *FID may not be reliable on DRealSR due to the limited number of test samples*. A small dataset can lead to unstable statistics in feature space, reducing the consistency and interpretability of FID. For this reason, we chose not to include FID results for DRealSR and RealSR in the main paper, as they may not reflect true perceptual performance.
>
> We thank you again for raising this important point. Your feedback has encouraged us to provide a more complete and nuanced evaluation of perceptual quality.

---

> ### Author Response · Authors · 2025-11-25
>
> > Question 1. The paper proposes “order classification” for real-world degradation modeling. I'm curious how effective this trick is and hope the authors can provide a ablation study or justification for it.
>
> **A. We are willing to provide further justification for "order classification".**
>
> As discussed in lines 300–307 of the previous maintext, in real-world scenarios, image degradation is often complex, mixed, and difficult to decompose into clearly defined categories. However, conventional "type-based" classification approaches, which are commonly used in all-in-one restoration tasks, rely on well-separated semantic labels. Because of this, type classification is ill-suited for the real-world super-resolution task.
>
> To address this limitation, we propose "order classification" as an alternative strategy. Instead of categorizing degradation by type, **"order classification" classifies samples based on the sequential order and cumulative complexity of degradation operations applied during the synthesis process.** For instance, in the degradation pipeline inspired by Real-ESRGAN, distortions are progressively introduced following the sequence:
> `+Blur -> +Resize -> +Noise -> +JPEG -> +Blur -> +Resize -> +Noise -> +JPEG`.
>
> Each step in this chain represents an increased level of degradation complexity. Accordingly, we define eight intermediate states (e.g., after the first operation, second, etc.), which serve as pseudo-labels indicating the stage (or "order") of degradation application. These orders act as surrogates for degradation severity and compositional complexity.
>
> By training the model to recognize these order levels, it learns to implicitly estimate how heavily an image has been degraded, enabling better adaptation to varying degrees of real-world distortion. This provides a more feasible and meaningful learning signal than attempting to assign discrete type labels that may not reflect actual conditions for real-world super-resolution task.
>
> ---
>
> As you have suggested, an ablation about type classification vs. order classification in real-world super resolution task would be ideal. However, due to the inherent ambiguity and overlap in real-world degradation types, *such alternatives cannot be reliably implemented or fairly evaluated under our setting*. Therefore, rather than comparing with impractical baselines, **we focus on justifying** the rationality and practical effectiveness of order classification in capturing degradation progression.
>
> ---
>
> Based on above analyzes, we acknowledge that the original description in lines 300-307 did not sufficiently explain the motivation and meaning of "order classification". **We have revised the corresponding part of the manuscript accordingly**, incorporating the above clarification to improve readability and conceptual transparency.
>
> - We added the above discussion of the "order classification" in lines 312-320.
>
> **Our revised PDF has been submitted.** In the revised PDF, the revised text is marked in blue.

---

> ### Author Response · Authors · 2025-11-25
>
> > Question 2. The paper mentions "full negative contrastive learning" in the RFT stage without defined what are negative or positive samples.
>
> **A.**  We sincerely appreciate the opportunity to clarify the meaning of "full negative contrastive learning". To provide clear context, we begin by reviewing the standard definitions of positive and negative samples in contrastive learning. Given an anchor sample,
>
> - **Positive samples** refer to instances that are semantically identical or highly related to the anchor sample.
> - **Negative samples** refer to those that are semantically different from the anchor.
>
> In the context of real-world image restoration addressed in this paper, these concepts are adapted based on degradation characteristics:
>
> - **Positive samples** are those sharing similar degradation types or levels of complexity as the anchor.
> - **Negative samples** are those exhibiting different degradation types or degrees of complexity.
>
> When training real-world super-resolution tasks, degradation is synthesized using Real-ESRGAN's degradation simulation pipeline, which involves a rich combination of diverse distortions such as blur, noise, compression, and downsampling. Due to the high variability in this synthesis process, even **within the same batch, different images are very likely to undergo distinct degradation patterns**. As a result, the probability of encountering truly matched degradation pairs (i.e., valid positives) is extremely low.
>
> Given this observation, we treat all samples within a batch as mutual negatives during learning, as we called it "full negative." This design choice reflects the practical reality of real-world degradation diversity and ensures stable and effective training under highly heterogeneous conditions.
>
> ---
>
> We acknowledge that the original description in lines 324–327 did not sufficiently explain the meaning of "full negative". We have revised the corresponding part of the manuscript accordingly, incorporating the above clarification to improve readability and conceptual transparency.
>
> - We added the above definition of the negative samples in lines 339-341.
>
> **Our revised PDF has been submitted.** In the revised PDF, the revised text is marked in blue.

---

### Meta-Review · Area_Chair_oAXR · 2026-01-07

**Summary:**

The paper introduces BDG (Bridging Degradation Discrimination and Generation), a framework for universal image restoration that tackles the dual challenge of identifying diverse image degradations and generating high-quality restorations. To address this, BDG integrates a fine-grained degradation characterization method called Multi-Angle and multi-Scale Gray Level Co-occurrence Matrix (MAS-GLCM), which outperforms prior techniques in distinguishing degradation types and levels. BDG’s training is structured into three stages: generation (to capture rich textures via diffusion), bridging (to align MAS-GLCM features with diffusion features), and restoration (to fine-tune fidelity while preserving discriminative capacity). This alignment enables the model to adaptively restore images across varied degradation scenarios without altering the underlying architecture.

**Reviewer Concerns:**

- The method is a minor extension of classical GLCM rather than a significant innovation.
Missing Comparisons: It lacks comparison against modern baselines like frequency-aware representations or learned degradation embeddings.
- The study omits crucial perceptual metrics (LPIPS, NIQE) and provides insufficient visual examples.
- There is no analysis of computational cost, memory usage, or inference speed.
- The three-stage training process with multiple loss functions is overly complex and difficult to tune.
- As a hand-crafted feature, it may fail on non-texture degradations (e.g., color shifts) compared to learned features.

**Reviewer Scores:**

Reviewer wiEb gave 6 before the rebuttal and requested some additional experiments and evalution metrics. The rebuttal shall solved the reviewer's concerns.

Reviewer ieqX gave 4 before the rebuttal. He/she didn't reply previous AC's inquiries. The succeeding AC ignores his/her comments.

Reviewer Rok7 gave 6 before the rebuttal. He/she mainly has concerns on missing comparison with simple features, missing details about computational cost, and the miss of other details. The rebuttal shall be able to solve the reviewer's concerns.

Reviewer JVFR  gave 6 before the rebutal. His/her concerns on insufficient visual results, missing experiments on unseen degradations, complex design and three-stage training shall be solved by the authors' rebuttal.

---

### Decision · Program_Chairs · 2026-01-26

Accept (Poster)